# Blocking ActRIIB and restoring appetite reverses cachexia and improves survival in mice with lung cancer

Andre Lima Queiroz[1,2,9], Ezequiel Dantas [1,2,9], Shakti Ramsamooj[1,2], Anirudh Murthy[1,2], Mujmmail Ahmed[1,2], Elizabeth R. M. Zunica [3], Roger J. Liang[1,2], Jessica Murphy[2,4,5], Corey D. Holman [6], Curtis J. Bare[6], Gregory Ghahramani [7], Zhidan Wu[8], David E. Cohen [6], John P. Kirwan[3], Lewis C. Cantley [2], Christopher L. Axelrod [3] & Marcus D. Goncalves [1,2] ✉

Cancer cachexia is a common, debilitating condition with limited therapeutic options. Using an established mouse model of lung cancer, we find that cachexia is characterized by reduced food intake, spontaneous activity, and energy expenditure accompanied by muscle metabolic dysfunction and atrophy. We identify Activin A as a purported driver of cachexia and treat with ActRIIB-Fc, a decoy ligand for TGF-β/activin family members, together with anamorelin (Ana), a ghrelin receptor agonist, to reverse muscle dysfunction and anorexia, respectively. Ana effectively increases food intake but only the combination of drugs increases lean mass, restores spontaneous activity, and improves overall survival. These beneficial effects are limited to female mice and are dependent on ovarian function. In agreement, high expression of Activin A in human lung adenocarcinoma correlates with unfavorable prognosis only in female patients, despite similar expression levels in both sexes. This study suggests that multimodal, sex-specific, therapies are needed to reverse cachexia.

Cancer cachexia is a highly prevalent wasting syndrome associated with progressive loss of skeletal muscle (with or without loss of fat mass) that predicts increased chemotherapy toxicity, complications from cancer surgery, and overall mortality[1,2]. Cachexia occurs commonly in patients with advanced lung cancer where the prevalence is estimated to be 45–60%[3,4]. Patients are diagnosed with cachexia when body weight drops more than 5% over the prior 6 months or more than 2% over 6 months if their body mass index (BMI) is already low (< 20 kg/m²) or if sarcopenia is present[5]. Cachexia induced weight loss is due to a combination of reduced food intake and hypermetabolism,

which arises from elevated energy expenditure, excess catabolism, and inflammation[5]. The therapeutic options for patients with cachexia are limited. For example, the latest clinical guidelines recommend no specific pharmacological interventions as the standard of care and millions of patients per year are left without treatment[6]. Therefore, there is an urgent need for pre-clinical discovery efforts that can be quickly translated into human interventions.

In rodent models of cachexia, tumor-secreted factors are considered the primary drivers of weight loss. Several tumor secreted proteins have been shown to directly bind to peripheral organs and

[1]Division of Endocrinology, Department of Medicine, Weill Cornell Medicine, New York, NY 10065, USA. [2]Meyer Cancer Center, Weill Cornell Medicine, New York, NY 10065, USA. [3]Pennington Biomedical Research Center, Baton Rouge, LA 70808, USA. [4]Center for Molecular Oncology, Memorial Sloan Kettering Cancer Center, New York, NY 10065, USA. [5]Department of Pathology, Memorial Sloan Kettering Cancer Center, New York, NY 10065, USA. [6]Division of Gastroenterology and Hepatology, Department of Medicine, Weill Cornell Medicine, New York, NY 10065, USA. [7]Weill Cornell Graduate School of Medical Sciences, Weill Cornell Medicine, New York, NY 10065, USA. [8]Internal Medicine Research Unit, Pfizer Global R&D, Cambridge, MA, USA. [9]These authors contributed equally: Andre Lima Queiroz, Ezequiel Dantas. ✉e-mail: mdg9010@med.cornell.edu

activate catabolic or suppress anabolic pathways to drive organ wasting[7–11]. For example, the Lewis Lung Carcinoma model of lung-cancer induced cachexia secretes parathyroid hormone-related protein and IL-17A leading to adipose tissue lipolysis and skeletal muscle atrophy, respectively[7,12].

Tumor secreted factors also promote anorexia[13–15]. For example, growth differentiation factor 15 (GDF-15) is produced by some tumors and activates anorexia via its receptor in the brainstem[14,16]. Anorexia reduces access to exogenous nutrients and exacerbates systemic metabolic dysfunction, and yet its contribution to weight loss has been underappreciated because it does not occur in the most frequently used pre-clinical models of cachexia[17,18]. In patients with advanced lung cancer, the prevalence of anorexia is as high as 66% so it is imperative that we identify models that feature this important co-morbidity[19]. To acknowledge the contributions of both cachexia and anorexia to weight loss during lung cancer progression, we will subsequently refer to the combined disorders as the cancer anorexia-cachexia syndrome (CACS).

We previously identified and characterized a genetic mouse model of lung cancer (KL: $Kras^{LSL-G12D/+};Lkb1^{flox/flox}$) that accurately reproduces CACS[20]. KL mice develop spontaneous, endogenous, aggressive lung tumors over several weeks following tumor induction[21]. Most, but not all, mice develop progressive weight loss, anorexia, and wasting of muscle and fat tissues. The incomplete penetrance of the CACS phenotype allows us to compare genetically identical, tumor-bearing mice with and without weight loss under controlled conditions. To date, there has not been a thorough profiling of the energetic changes that occur in KL mice with and without CACS. Such an analysis may reveal therapeutic targets and interventions that improve the lives of patients with lung cancer.

During CACS, anorexia and increased energy expenditure (EE) may contribute to the state of negative energy balance that leads to weight loss. Total energy expenditure (TEE) is controlled centrally with input from peripherally derived hormones like leptin, thyroid hormones, and glucocorticoids[22–24]. TEE can be divided into activity energy expenditure (AEE) and resting energy expenditure (REE)[25]. Intuitively, animals and patients with lung cancer should have increased REE given the presence of highly metabolic tumor cells, however, the data supporting this assumption are mixed[26].

Skeletal muscle is a primary determinant of both AEE and REE given its large mass, ability to mediate locomotion, and the substantial capacity for mitochondrial respiration[27–29]. In cachexia, skeletal muscle mitochondrial dysfunction is a hallmark feature observed across clinical samples and mouse models including KL mice[20,30]. Therefore, it is likely that alterations in skeletal muscle contribute to the changes in energy balance during CACS.

In this study, we used the KL model to interrogate the changes in food intake, peripheral organ metabolism, and EE that occur following the induction of lung cancer. We found that mice with CACS have severely low TEE driven by reductions in food intake, spontaneous activity, and altered skeletal muscle metabolism. These changes correlated with high levels of tumor-derived Activin A and activation of the TGF-β/SMAD transcriptional program in the muscle. Based on this data, we hypothesized that CACS could be reversed by targeting both anorexia and Activin A-induced metabolic dysfunction. KL mice were treated with a ligand trap for TGF-β/activin family members (ActRIIB-Fc) in combination with agents that improve appetite (anti-GDF-15 antibody and anamorelin (Ana), a ghrelin receptor agonist). We found that only Ana increased food intake and fat mass, and the combination of Ana and ActRIIB-Fc further increased lean mass, restored spontaneous activity, and improved overall survival. These beneficial effects were limited to female mice and were dependent on ovarian function. In agreement, we found that high expression of Activin A in human lung adenocarcinoma correlates with unfavorable prognosis only in female patients, even though expression levels are similar in both sexes. Overall, our study suggests that multimodal, sex-specific, therapies are needed to reverse CACS.

## Results

### Food intake and energy expenditure are reduced during CACS

Following induction with an inhaled adenovirus carrying Cre recombinase (AdCre), a large proportion of KL mice (~70%) develop CACS, defined as >15% body weight loss (Fig. 1a, b)[20]. The remaining mice, classified as non-cachectic (-CACS), do not reach this threshold despite similar tumor burden (Fig. 1c). The body weight lost during CACS is due to a reduction of both lean and fat mass that occurs concomitantly at a late stage (~7–8 weeks after induction) (Fig. 1d, e). The reduction in total body fat is reflected in the mass of the gonadal white adipose tissue (gWAT) and interscapular brown adipose tissue (BAT) depots, which both decrease in mass linearly as total weight is reduced (Fig. S1a). The reduction in gWAT mass in mice with CACS is associated with dramatic adipocyte atrophy, increased rates of triglyceride release, and higher levels of non-esterified fatty acids (NEFA) in the serum, suggestive of ongoing lipolysis (Fig. S1b–e). Smaller adipocytes are known to suppress the production of leptin, an adipokine that regulates food intake and EE, and we confirmed this finding in cachectic KL mice at the mRNA level and in the serum (Fig. S1f, g). Other factors that regulate EE like catecholamines, thyroid hormones, and insulin were unchanged (Table 1). As we previously reported, corticosterone levels were higher in mice with CACS[20].

Previous reports suggest that 'browning' of the WAT is an important contributor to EE in CACS[7,31]. Therefore, we assessed the expression of genes related to browning in the gWAT of KL mice with and without CACS using BATLAS, an algorithm capable of estimating the fraction of brown adipocyte content using RNA-Seq expression data[32]. The results demonstrate that cachectic mice have higher rates of browning, as compared to tumor-bearing mice without CACS (Fig. S1h). We confirmed this result by measuring the abundance of UCP1 protein by immunostaining and mRNA expression of $Ucp1$ and $Ppargc1a$ in the gWAT (Fig. S1i–k). Of note, we were unable to detect UCP1 protein in the gWAT by Western blot.

UCP1 is primarily expressed in the brown adipose tissue (BAT) where it plays an important role in thermogenesis. To determine the impact of CACS on the BAT, we profiled this tissue histologically. We found that both groups of tumor-bearing mice (-CACS and +CACS) had increased lipid content in the BAT (Fig. S1l, m), a phenotype that has been associated with defective glucose uptake and thermogenesis in rodent models of obesity and diabetes[33,34]. While there was no statistically significant difference in the mean intensity of UCP1 immunohistochemistry, $Ucp1$ mRNA levels were significantly decreased in the cachectic BAT as compared to tissue from mice without CACS (Fig. S1l–o). In line with a potential loss of BAT function, we found lower core temperatures in mice with CACS (Fig. S1p).

Mice need to actively generate heat to maintain body temperature under standard housing conditions (22 °C). This adaptive thermogenesis increases REE and may disproportionately contribute to weight loss during CACS. To assess the contribution of cold-induced thermogenesis to CACS, we performed a prospective, randomized, controlled trial (RCT) where KL mice were induced with AdCre at 22 °C and then randomly assigned to stay at 22 °C or move to a thermoneutral temperature (30 °C) four weeks after induction, which allows enough time for normal tumor development. When mice are housed at 30 °C, there is minimal contribution of cold-induced thermogenesis to total EE[35,36]. The thermoneutral environment did not alter the pattern of weight loss, adipose tissue wasting, total BAT mass, muscle mass, nor did it change overall survival (Fig. S2a–e). We were unable to detect any histologic evidence of browning in the cachectic gWAT when mice were housed at 30 °C; however, the number of mice with available tissue was small ($n = 5$ per group) and this limited our ability to make a definitive conclusion (data not shown). Overall, these results suggest

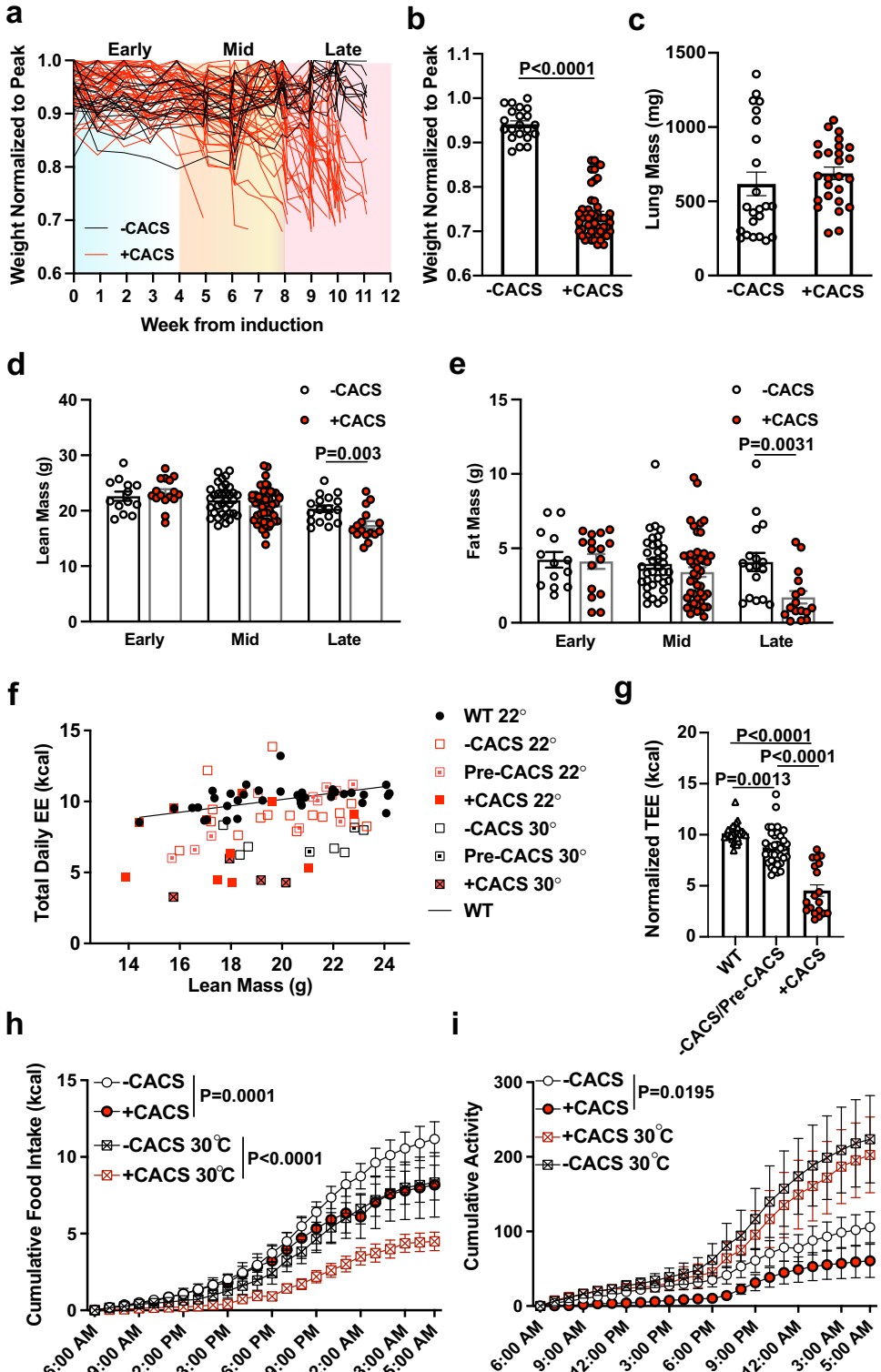

that gWAT browning and cold-induced thermogenesis do not contribute to CACS in this model.

During the thermoneutral experiment, we observed that the EE of mice with CACS was much lower than mice without CACS at both temperatures, with oxygen consumption (VO2) following a similar trend (Fig. S2f, g). Mice with CACS also had lower values of the respiratory exchange ratio (RER) as compared to those without CACS at both temperatures (Fig. S2h). RER is an indirect readout of whole-body substrate oxidation and lower values are consistent with less glucose or ketone oxidation, a finding we previously described[20].

We plotted the TEE values against lean mass to compare the values to KL littermates that never received tumor induction (WT), tumor-bearing mice that did not lose weight (-CACS), and tumor-bearing mice that were weight-stable at the time of the measurement but later developed CACS (referred to as Pre-CACS) (Fig. 1f, g). Interestingly, we observed a stepwise reduction in TEE with WT values being the highest, and CACS (+CACS) being the lowest. Pre-CACS and tumor-bearing weight stable (-CACS) mice had intermediate values. We confirmed this observation by normalizing TEE to lean mass using ANCOVA, as previously described (Fig. 1g)[36,37]. In addition to the low

**Fig. 1 | Appetite and energy expenditure are reduced during CACS.**
**a** Progression of weights normalized to their peak since tumor induction ($n = 70$).
**b** The normalized weight of the mice in A at endpoint. **c** Lung mass of mice with and without CACS at endpoint (-CACS $n = 23$, +CACS $n = 26$). Lean (**d**) and Fat mass (**e**) measured 4 weeks after induction, defined as "Early",(-CACS $n = 13$, +CACS $n = 16$), 4 to 8 weeks from induction defined as "Mid", (-CACS $n = 35$, +CACS $n = 48$), and before euthanasia, defined as "Late" stages of disease, (-CACS $n = 17$, +CACS $n = 16$). **f** Total daily energy expenditure versus total body lean mass for WT ($n = 29$), -CACS, tumor-bearing mice without weight loss that later developed CACS (Pre-CACS), ($n = 35$), and +CACS ($n = 19$) either at 22 °C or 30 °C. The black line corresponds to WT mice at 22 °C ($n = 29$), Pre-CACS and - CACS mice are represented by dotted red squares ($n = 35$) and +CACS mice (22 °C and 30 °C, $n = 19$) by solid red squares. **g** Total daily energy expenditure adjusted by lean mass using ANCOVA using data from **f**. Cumulative food intake (**h**) in kcal and Cumulative activity (**i**) (distance traveled in meters), over a representative 24 h period at 22 °C of -CACS ($n = 15$) and + CACS ($n = 5$) or at 30 °C of -CACS ($n = 8$) and + CACS mice ($n = 8$). Both male and female mice were used in all panels. Graphs **b**–**e**, **g**–**i** show mean ± SEM. Comparisons in **b**–**e** were made with two-tailed Student's $t$-test between with +CACS and -CACS. Comparisons in **g** were done using one-way ANOVA followed by Tukey's multiple comparisons test. Comparisons in **h**, **i** were made by 2-way ANOVA followed by Tukey's multiple comparisons test. Individual data points are independent biological replicates unless otherwise stated. Source data are provided as a Source Data file.

TEE, the mice with CACS had lower food intake and spontaneous activity as compared to mice without CACS, however the effect on activity faded when mice were housed at thermoneutrality (Fig. 1h, i).

These data reveal that KL mice with CACS have anorexia and reduced TEE, despite increased biomarkers of WAT browning. Since skeletal muscle is the primary contributor to REE and physical activity in humans and rodents[27–29], we hypothesized that the overall hypometabolic state was due to muscle metabolic dysfunction.

### Mice with CACS have impaired skeletal muscle metabolism

While TEE is a function of lean mass (mostly skeletal muscle), the reduction in TEE during CACS is disproportionally lower than what is predicted by the lower lean mass (Fig. 1e). A similar observation has been made in obese humans undergoing intentional weight loss[38]. In this setting, there is a decline in skeletal muscle glycolytic activity, which increases work efficiency and lowers EE[39]. We previously demonstrated and now confirm that the reduction in skeletal muscle mass in cachectic KL mice is specific to muscles containing glycolytic, fast-twitch fibers such as the EDL, gastrocnemius, and quadriceps, while muscles containing oxidative, slow twitch fibers such as soleus are preserved (Fig. S3a, b). To assess glycolytic activity, we measured the activity of phosphofructokinase (PFK) in glycolytic (EDL) and oxidative (Soleus) skeletal muscle lysates derived from mice with and without CACS. As expected, PFK activity was higher in the EDL as compared to Soleus, and it was lowered by CACS only in the EDL

#### Table 1 | Circulating Hormone Levels

| Hormones (ng/ml) | Mean (SEM) |
|---|---|
| **Corticosterone** | |
| -CACS | 38.7 (5.5) |
| +CACS | 227.0 (59.5)* |
| **Norepinephrine** | |
| -CACS | 26.10 (2.13) |
| +CACS | 31.02 (2.40) |
| **Epinephrine** | |
| -CACS | 61.72 (8.53) |
| +CACS | 45.51 (5.23) |
| **T3** | |
| -CACS | 0.77 (0.06) |
| +CACS | 0.73 (0.12) |
| **T4** | |
| -CACS | 27.85 (2.22) |
| +CACS | 30.39 (5.61) |
| **Insulin** | |
| -CACS | 0.85 (0.08) |
| +CACS | 0.71 (0.10) |

*$p = 0.0033$.
Mean ± SEM of the serum concentration of corticosterone, norepinephrine, epinephrine, t3, t4, and insulin in -CACS and +CACS mice. Comparisons were made using two-tailed Student's $t$-test.

(Fig. 2a). We also surveyed the activity of other metabolic enzymes using phosphorylation-based biomarkers that correlate with enzyme activity. For example, the phosphorylation of LDH-A at Tyrosine 10 correlates with higher activity of the enzyme[40]. This modification was increased in the EDL but not the Soleus of mice with CACS (Fig. 2b–d). In addition, the phosphorylation of the E1α subunit of PDH at Serine 293 is associated with lower enzymatic activity[41], and this biomarker was also increased specifically in the EDL of mice with CACS (Fig. 2b–d). Together, these results suggest an overall reduction in glucose oxidation in fast-twitch muscles.

We performed RNA-seq using the gastrocnemius to identify transcriptional signatures that could explain the metabolic and phenotypic changes observed in the muscles of the mice with CACS. At a whole-transcriptome level, mice with and without CACS clustered independently in an unbiased principal component analysis (PCA) (Fig. S3c). Interestingly, a Gene Set Enrichment Analysis (GSEA) revealed that the muscles from the cachectic mice had decreased mitochondrial function and TCA cycle activity (Fig. S3d, e). These observations were confirmed by western blot where we found decreased levels of mitochondrial complex I and II in the EDL of CACS mice (Fig. 2e–g). Furthermore, we found that the mitochondrial DNA content, a surrogate for mitochondrial abundance, was reduced in these muscles (Fig. 2h). However, there were no changes in muscle citrate synthase activity nor the oxidative phosphorylation capacity of permeabilized soleus and EDL fibers (Fig. S3f–h). Thyroid hormones are known to alter skeletal muscle metabolism during weight loss but there was no change in the abundance of T3 and T4 in muscle extracts (Fig. S3i).

To test the function of skeletal muscle in vivo, we performed a maximal exercise capacity test using a motorized treadmill. The total distance traveled, time until exhaustion, and work performed significantly correlated with weight loss demonstrating a reduction in exercise capacity in mice with CACS (Fig. 2i). Blood lactate levels after exercise were similar in both groups suggesting that the mice with CACS reached their lactate threshold at a lower workload (Fig. 2j), an expected result in the setting of mitochondrial dysfunction[42]. Combined with the biochemical assessment of skeletal muscle enzyme activity, these results suggest that glucose-derived carbon is being diverted away from the TCA cycle and oxidative phosphorylation. Indeed, the steady-state abundance of the TCA intermediates citrate, fumarate, and malate were reduced (Fig. 2k–m). These studies reveal deficits in oxidative phosphorylation that occur in the glycolytic skeletal muscles of mice with CACS that may limit spontaneous activity, forced exercise capacity, and TEE.

### Caloric Restriction does not recapitulate the metabolic changes observed in CACS

Anorexia reduces TEE and is a confounding factor when trying to understand the energetics of CACS[43,44]. To assess the contribution of reduced food intake to TEE and skeletal muscle metabolism, we performed an experiment where WT mice were calorie-restricted (CR) to consume the same energy as mice with CACS (8 kcal/day as shown in

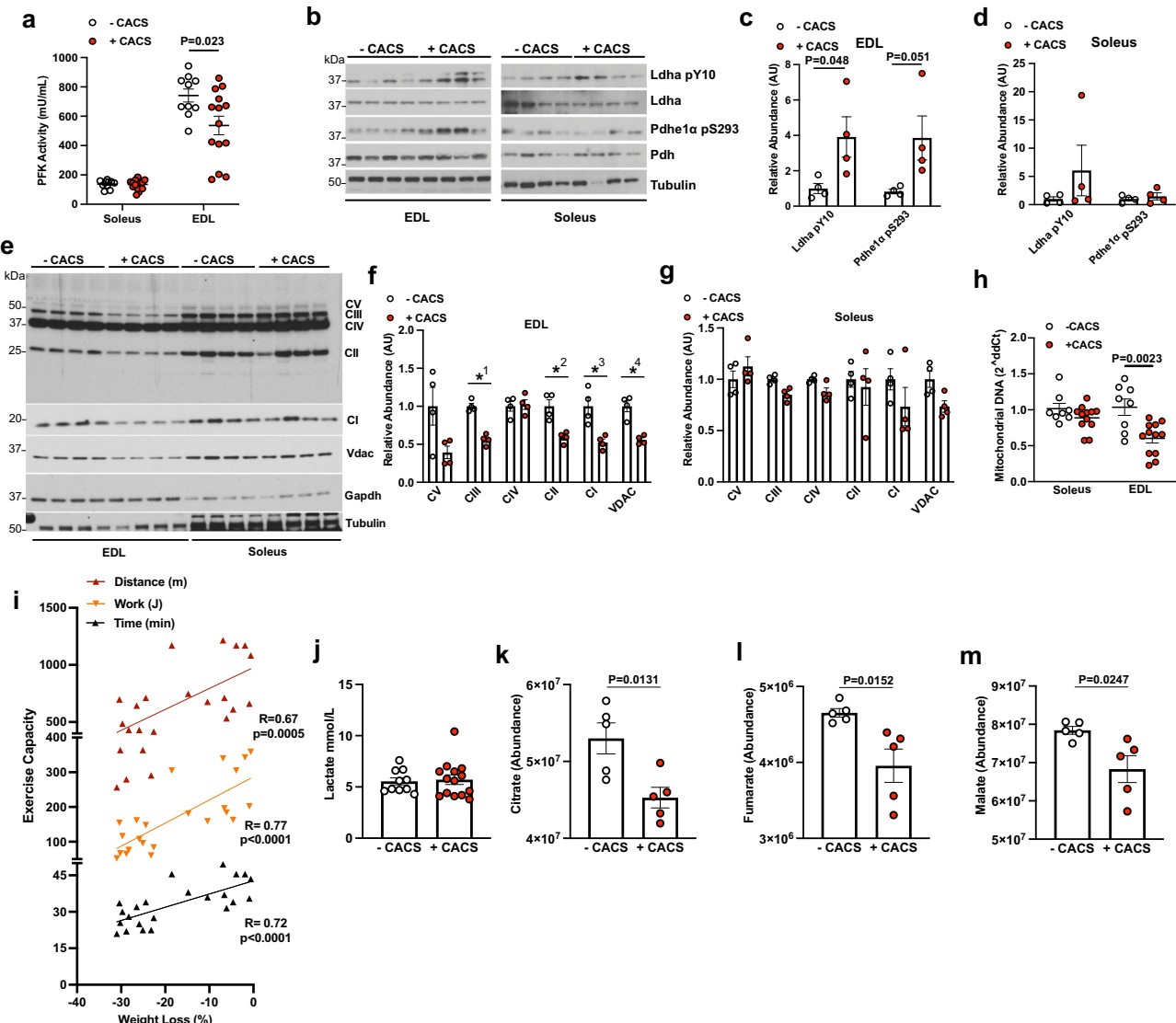

**Fig. 2 | Mice with CACS have impaired skeletal muscle metabolism.**
**a** Phosphofructokinase (PFK) activity measured using Soleus lysates from mice without (-CACS, $n = 10$) and with cachexia (+CACS, $n = 15$) and extensor digitorum longus (EDL) lysates from mice without (-CACS, $n = 11$) and with cachexia (+CACS, $n = 15$). **b** Western blot of phosphorylated (Tyr10) and total LDHa, phosphorylated (Ser293) and total PDHe1α, and Tubulin from Soleus and EDL lysates from -CACS and +CACS mice. **c** Relative quantification of WB shown in B for phosphorylated (Tyr10) LDHa and phosphorylated (Ser293) PDHe1α in EDL lysates from -CACS ($n = 4$), and +CACS ($n = 4$)mice. **d** Relative quantification of WB shown in B for phosphorylated (Tyr10) LDHa and phosphorylated (Ser293) PDHe1α in Soleus lysates from -CACS ($n = 4$), and +CACS ($n = 4$) mice. **e** Western blot analysis of mitochondrial oxidative phosphorylation complexes (CI-subunit NDUFB8, CII-SDHB, CIII-UQCRC2, CIV-MTCO1, and CV-ATP5A) as well as VDAC, GAPDH, and Tubulin in lysates of Soleus and EDL muscles from -CACS and +CACS mice.
**f** Relative quantification of WB shown in E for mitochondrial oxidative phosphorylation complexes in EDL lysates from -CACS ($n = 4$), and +CACS ($n = 4$)

mice($P$-value = *1 = 0.0001, *2 < 0.004, *3 < 0.007, *4 < 0.001). **g** Relative quantification of WB shown in E for mitochondrial oxidative phosphorylation complexes in Soleus lysates from -CACS ($n = 4$), and +CACS ($n = 4$) mice. **h** Relative mitochondrial DNA content in Soleus and EDL muscles of -CACS($n = 8$) and +CACS($n = 8$) mice. **i** Distance traveled (m), work performed (J), and duration (min) of maximal endurance performance test (running on a treadmill until exhaustion) of KL mice versus total body weight loss. Linear regression of each metric is shown. **j** Blood lactate levels of -CACS ($n = 10$) and +CACS($n = 14$) mice at the completion of the maximal endurance performance test in l. **k** Citrate, (**l**) Fumarate, (**m**) Malate levels in gastrocnemius extracts from -CACS ($n = 5$) and +CACS($n = 5$) mice by mass spectrometry. Both male and female mice were used in all panels. Graphs show mean ± SEM. **a**, **c**, **d**, **f–h**, **j–m** comparisons were made using two-tailed Student's $t$-test compared with -CACS mice. Comparison in **i** was made using correlation analysis (R, Pearson r, and p, $P$-value). Individual data points are independent biological replicates unless otherwise stated. Source data are provided as a Source Data file.

Fig. 1h). CR led to similar changes in weight, body composition, and TEE as CACS (Fig. 3a–c). In contrast to CACS, CR induced a 4-fold increase in spontaneous activity, which has been previously described as "food-seeking behavior" (Fig. 3d)[45]. We used the TEE and activity data to estimate the REE and AEE of WT, tumor-bearing, and CR mice using a penalized spline regression (Fig. S4a, b)[46]. From this analysis, we observed significant reductions in REE in the CACS and CR mice in comparison to WT mice and mice without CACS, however no

difference between CACS and CR. The increase in spontaneous activity in the CR mice led to a significant change in AEE between CR and CACS. To assess skeletal muscle work efficiency, we estimated the caloric cost of activity (CCA) using AEE and physical activity data. There was a trend for the CCA to be lower in CR mice as compared to WT, suggestive of improved work efficiency (Fig S4c)[47,48]. In agreement, CR mice could travel longer distances during an exercise performance test, despite performing the same amount of work (Fig. 3e, f).

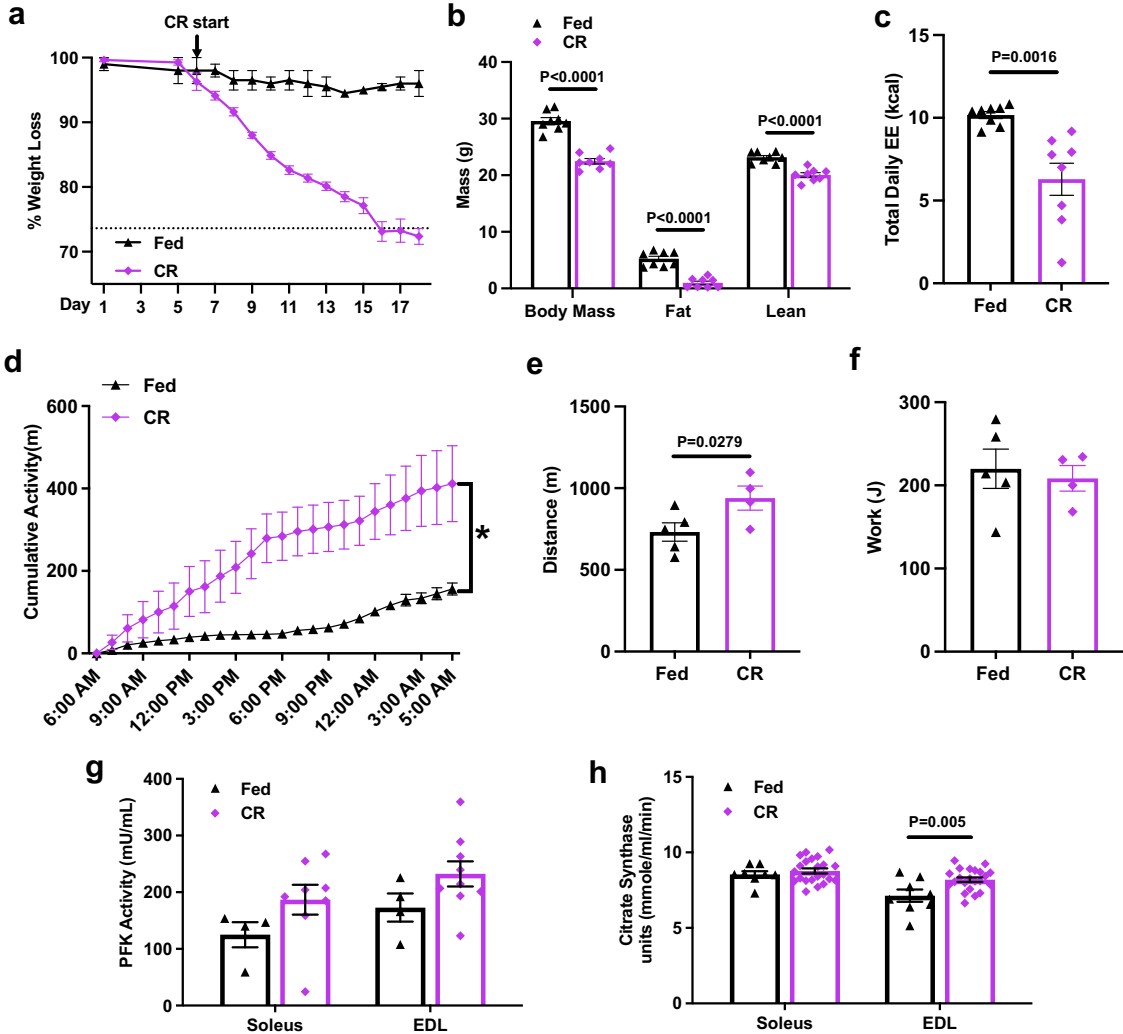

**Fig. 3 | Caloric Restriction does not recapitulate the metabolic changes observed in CACS.** Wild-type mice were calorie restricted (CR) by feeding a ~8 kcal/day diet (amount consumed by cachectic mice) until weight stabilization. **a** Percent of body weight change over a period of 18 days of CR ($n = 8$), or control diet (Fed, $n = 2$); The dashed line on the y-axis reflects the mean calorie consumption of mice with CACS taken from Fig. 1b. **b** Total body weight, Fat, and Lean mass of Fed ($n = 8$) and CR($n = 8$) mice of **a** at day 18. **c** Total daily energy expenditure (kcal) in 24 h of Fed ($n = 8$) and CR($n = 8$). **d** Cumulative activity (distance traveled in meters) over a representative 24 h period of Fed($n = 8$) and CR mice($n = 8$). **e** Distance traveled (m) and (F) Work (J) performed by Fed($n = 5$) and CR ($n = 4$) mice performing a maximal endurance performance test (running on a treadmill until exhaustion). **g** Phosphofructokinase (PFK) activity in the soleus of Fed ($n = 4$) and CR mice ($n = 8$) and in the EDL($n = 4$) of Fed and CR mice($n = 9$). **h** Citrate Synthase (CS) activity in the soleus of Fed ($n = 8$) and CR mice ($n = 22$) and in the EDL($n = 8$) of Fed and CR mice($n = 22$). All mice used in the figure were wild-type males. Graphs show mean ± SEM. **a–c**, **f–h** comparisons made using two-tailed Student's $t$-test compared with -CACS mice, (**e**) with a one-tailed Student's $t$-test and **d** by 2-way ANOVA with Tukey's multiple comparisons. Individual data points are independent biological replicates unless o therwise stated. Source data are provided as a Source Data file.

Similar to CACS, CR reduced the mass of the gWAT, BAT, and glycolytic (but not oxidative) skeletal muscles (Fig S4d, e). Histologically, the gWAT and skeletal muscle displayed atrophy with small adipocytes and reduced fiber cross-sectional area, respectively (Fig S4f–i). Contrary to what was observed in mice with CACS, the BAT of CR mice was depleted of lipid and the UCP1 tissue staining was more intense (Fig S4f, h, j–l). In the gWAT, UCP1 positive cells were only rarely identified (Fig S5j, k).

Although the skeletal muscle from mice with CACS and CR shared signs of atrophy, the muscles from the latter did not suffer the same metabolic and mitochondrial dysfunction. The PFK activity tended to increase in both the Soleus and EDL, and the citrate synthase activity was significantly higher in the EDL muscles of the CR mice (Fig. 3g, h). Furthermore, no changes in the abundance of the electron transport chain proteins were observed in the muscles of the CR mice (Fig. S4m, n).

Overall, these data reveal CACS-specific perturbations in physical activity and peripheral organ metabolism that occur independently of reduced food intake, body composition, and lower TEE.

## Activin A is a Tumor-secreted Factor that Correlates with CACS in KL mice

To identify tumor secreted factors responsible for the CACS-specific metabolic perturbations, we performed RNA-Seq using tumors from KL mice with and without CACS. At a whole-transcriptome level, the tumors from each group clustered independently in an unbiased PCA (Fig. 4a). Among the most differentially expressed genes (DEG), we found *Inhba*, the gene coding for Activin A, a member of the TGF-β superfamily capable of inducing muscle atrophy and modulating adipocyte browning (Fig. 4b)[49–51]. This finding agreed with the GSEA of the skeletal muscles that revealed activation of TGF-β/SMAD as one of the most enriched pathways (Fig. S3e).

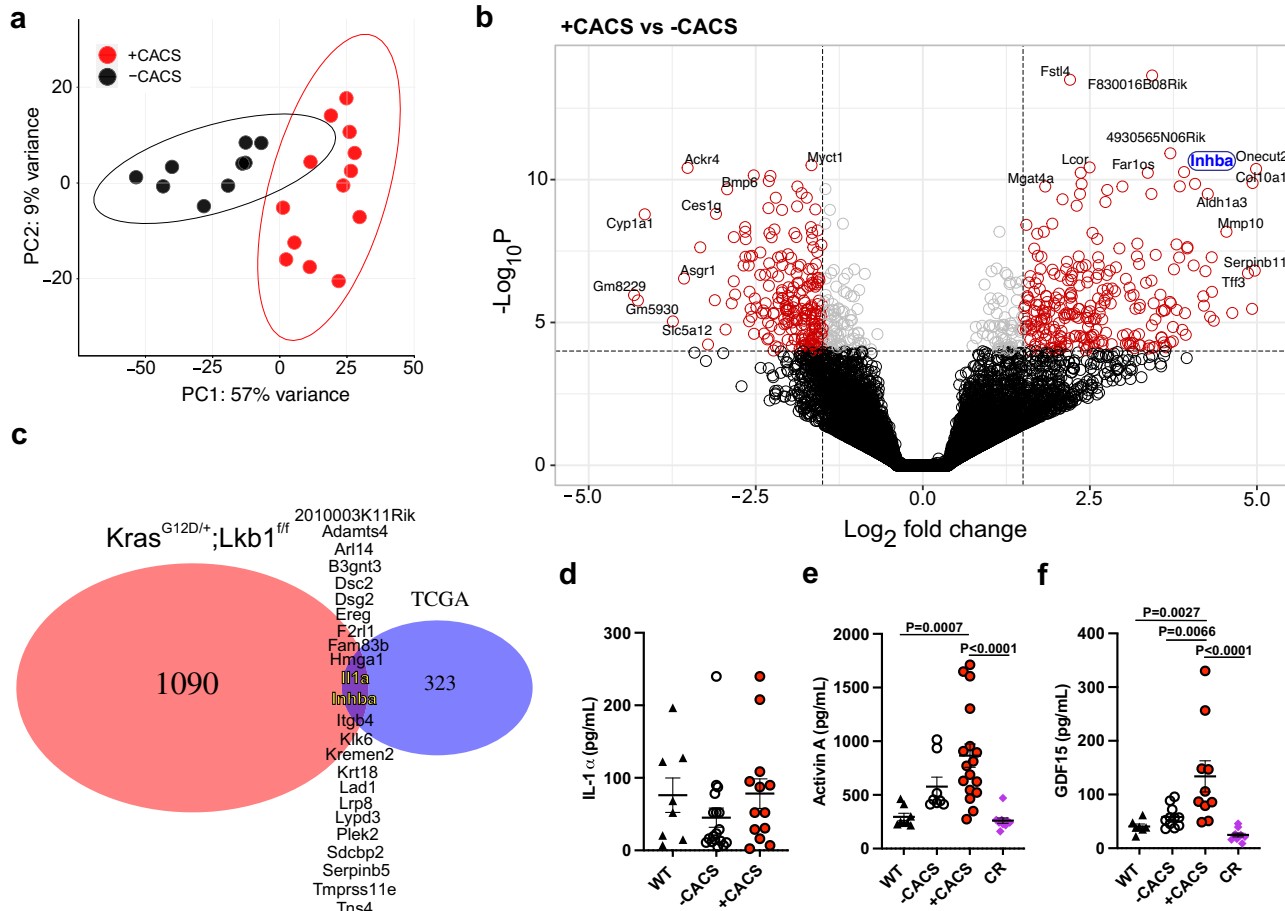

**Fig. 4 | Tumors from mice with CACS have higher Activin A expression.**
**a** Principal Component Analysis (PCA) of RNA-Seq from tumors of -CACS ($n = 9$) and +CACS ($n = 13$) mice. **b** Volcano plot depicting INHBA as one of the most differentially expressed genes (DEG) in the RNA-Seq of tumors from +CACS mice compared to -CACS ($P < 0.01$ & $Log_2$Fold Change >1.2). **c** Venn diagram between the DEG identified in **b** and the human genes associated with poor prognosis in lung cancer from TCGA. **d** IL-1α levels in the serum of Fed ($n = 8$), -CACS ($n = 10$) and +CACS ($n = 14$) mice at the time of euthanasia. **e** Activin A levels in the serum of Fed ($n = 8$), -CACS ($n = 8$), +CACS ($n = 17$) and CR ($n = 10$) mice at the time of euthanasia. **f** GDF15 levels in the serum of Fed ($n = 7$), -CACS ($n = 11$), +CACS ($n = 10$), and CR ($n = 10$) mice at the time of euthanasia. **a**, **b** were done with samples from male mice, serum measurements in **d**–**f** were performed in both male and female mice. Graphs show mean ± SEM. **d**–**f** comparisons were made using One-way ANOVA with Tukey's multiple comparison test. Individual data points are independent biological replicates unless otherwise stated. Source data are provided as a Source Data file.

To assess the clinical relevance of this finding, we compared the KL tumor DEGs to a list of 345 human lung cancer DEGs associated with poor prognosis from the TCGA[52]. Of the 24 genes that matched between both groups (Fig. 4c), the only 'secreted' genes by the Gene Ontology (GO) database were *Il1a* (encoding for IL-1α) and *Inhba*. We measured IL-1α and Activin A in the serum and found that only the latter was significantly increased in KL mice with CACS (Fig. 4d, e).

No candidate genes were found that could explain the anorexic phenotype of KL mice, so we performed a targeted serum analysis for relevant candidates. In other cachexia models, GDF-15 is capable of inducing anorexia through the activation of GFRAL in the brainstem[14,16]. We found high levels of GDF-15 in KL mice with CACS compared with WT, CR, and tumor bearing mice without CACS (Fig. 4f). These data suggest that GDF-15 and Activin A are viable targets for treating CACS in KL mice.

### Targeting GDF-15 and Activin A does not reverse CACS

We performed a prospective RCT with ActRIIB-Fc, a decoy ligand for TGF-β/activin family members like Activin A, and a monoclonal antibody (mAb) targeting GDF-15 to determine the effects of this combination therapy on food intake, TEE, and survival. Mice were induced with AdCre and then monitored weekly for changes in body weight and food intake. Once the mice reached 15% weight loss, they were

randomized to one of the three intervention arms (control, anti-GDF-15 mAb, and the combination of ActRIIB-Fc and anti-GDF-15 mAb). By using this approach, our goal was to treat existing CACS instead of preventing the onset of CACS. Surprisingly, neither the anti-GDF-15 mAb or combination therapy increased food intake, and only the mice treated with the combination therapy showed a significant attenuation of weight loss (Fig. 5a, b). This response was driven by two female mice who displayed modest weight regain. The combination therapy did not change fat mass and led to a subtle improvement in lean mass (Fig. 5c, d), with no effects on overall survival (Fig. 5e), tumor burden, spontaneous activity, RER, or EE (Fig. S5a–d).

### Anamorelin and ActRIIB-Fc combination treatment improves weight, activity, and overall survival in female KL mice with CACS

Because the anti-GDF-15 mAb did not improve food intake as expected, we searched for other therapies that would reverse anorexia. Anamorelin (Ana) was recently approved by the pharmaceutical regulatory authority of Japan for the treatment of patients with CACS[53]. Ana is a ghrelin receptor agonist that has been reported to induce food intake and improve whole body weight and lean mass in patients with CACS[54–56]. Therefore, we planned a prospective RCT in KL mice using Ana and ActRIIB-Fc. Using a similar design as the GDF-15 trial, we

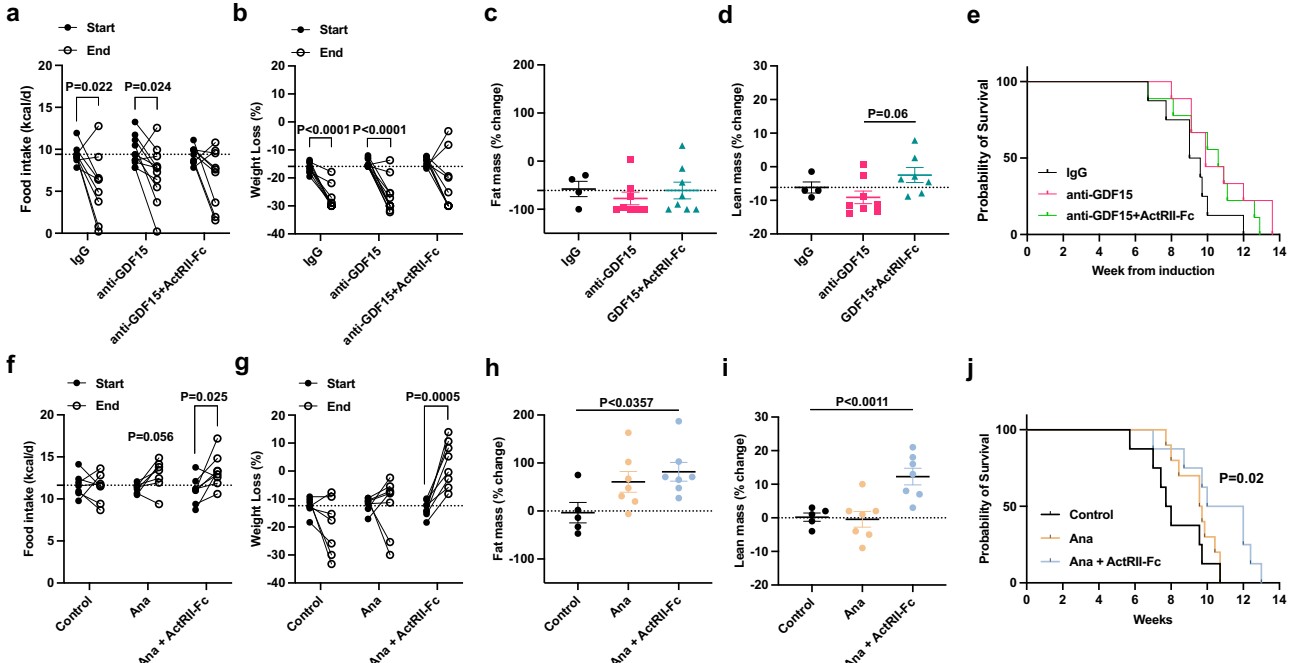

**Fig. 5 | Anamorelin but not GDF15 in combination with ActRIIB-Fc improves body weight, activity, and overall survival in female KL mice with CACS. a** Food intake (kcal/d) at week 0 (Start) and after 2 weeks of treatment (End) with control immunoglobulin (IgG, $n = 9$), anti-GDF15 mAB ($n = 11$) or ActRIIB-Fc decoy mAb in combination with GDF15 mAb ($n = 9$). **b** Total body weight loss at Start and End in mice treated with either IgG ($n = 9$), anti-GDF15 monoclonal antibody ($n = 11$) or anti-GDF15 mAb together with an ActRIIB-Fc decoy mAb ($n = 9$). Percentage of fat mass (**c**) and lean mass (**d**) change after 2 weeks of treatment with either IgG ($n = 4$), anti-GDF15 monoclonal antibody ($n = 8$) or anti-GDF15 mAb together with a decoy ActRIIB-Fc mAb ($n = 8$). **e** Kaplan-Meier (KM) plot with the probability of survival of the mice from (**b**). **f** Food intake (kcal/d) at Start and End with control IgG ($n = 7$), anamorelin (Ana, $n = 7$) or ActRIIB-Fc decoy mAb in combination with Ana ($n = 8$). **g** Total body weight loss at Start and End in mice treated with either

IgG ($n = 7$), Ana ($n = 8$) or Ana in combination with ActRIIB-Fc decoy mAb ($n = 8$). **h** Percentage of fat mass change after 2 weeks of treatment with either IgG ($n = 5$), Ana ($n = 7$) or Ana in combination with ActRIIB-Fc decoy mAb ($n = 7$). **i** Percentage of lean mass change after 2 weeks of treatment with either IgG ($n = 5$), Ana ($n = 6$) or Ana in combination with ActRIIB-Fc decoy mAb ($n = 7$). **j** Kaplan-Meier (KM) plot with the probability of survival of the mice from **g**. **a–e** Both male and female mice were used; (**f–j**) only female mice were used. Graphs show mean ± SEM. **a, b, f, g** comparisons were made using two-tailed paired Student's *t*-test comparing "Start" to "End" for each treatment. Comparisons in **c, d, h, i** were done using one-way ANOVA followed by Tukey's multiple comparisons test. Comparisons in (**e, j**) were made using Log-rank Mantel-Cox test. Individual data points are independent biological replicates unless otherwise stated. Source data are provided as a Source Data file.

randomized KL mice into three intervention arms (Control, Ana, or Ana and ActRIIB-Fc). Mice treated with Ana showed trends toward improvement in food intake and fat mass, but only the combination therapy arm reversed weight loss and improved fat and lean mass (Fig. S6a–d). There were no significant changes in overall survival and EE in either treatment arm (Fig. S6e, f); however, spontaneous activity was increased in the mice treated with Ana and ActRIIB-Fc (Fig. S6g).

During this trial, we noticed that female mice seemed to respond better than male mice. Therefore, we analyzed the cohort by sex and interrogated the therapeutic efficacy of these interventions. In this analysis, it was clear that male mice did not benefit from either intervention (Fig. S6h–k). However, there was a strong effect in female mice. Ana showed a trend towards increased food intake, weight, and fat mass in the female mice (Fig. 5f–h). The addition of ActRIIB-Fc to Ana further improved weight due to an increase in lean mass (Fig. 5i). Remarkably, CACS was fully reversed in all the female mice treated with the combination of Ana and ActRIIB-Fc (Fig. 5g) without changes in lung mass, a surrogate for tumor burden (Fig. S5e). The combination therapy also restored spontaneous activity (Fig. S5f) and improved overall survival (Fig. 5j) without changes to RER or EE (Fig. S5g–h).

There is a limited understanding of the basic mechanisms underlying sex differences in CACS[57]. Activin A levels were not appreciably different between male and female mice (Fig S7a). To test if the response to treatment was estrogen dependent, we ran a new RCT where only female mice were randomized to three arms: control, fulvestrant (an estrogen receptor degrader), or fulvestrant and the

Ana/ActRIIB-Fc combination therapy. Fulvestrant had no effect on the therapeutic response to the combination therapy, suggesting that peripheral estrogen receptor activity does not play a role (Fig. S7b–e). Since the ovary produces other hormones that could be responsible for the therapeutic response of the combination therapy, we ran another RCT where female mice underwent ovariectomy or sham surgery prior to tumor induction and were then randomized to receive the combination therapy or vehicle at the onset of CACS. As expected, the combination therapy tended to improve food intake, weight, and body composition (Fig. S7f–i). In contrast, the ovariectomized mice treated with the combination therapy only showed trends towards higher lean mass with no improvements in food intake, weight loss, or fat mass. We conclude that the beneficial effects of the combination therapy on female mice with CACS depends on an ovarian factor that does not signal through the estrogen receptor.

**Activin A expression in human lung adenocarcinoma is associated with poor prognosis only in female patients**

To assess the clinical relevance of our findings, we interrogated data from the Clinical Proteomic Tumor Analysis Consortium (CPTAC), which includes tumor proteomics data and clinical features such as the patient's body mass index (BMI)[58]. Using a cutoff of BMI ≤ 20 kg/m$^2$ to approximate CACS[2], we confirm that protein levels of Activin A are higher in the tumors from patients with cachexia (Fig. 6a). In order to exert its metabolic effects on muscle, Activin A should also be found in the serum so we probed for this protein in an independent cohort of

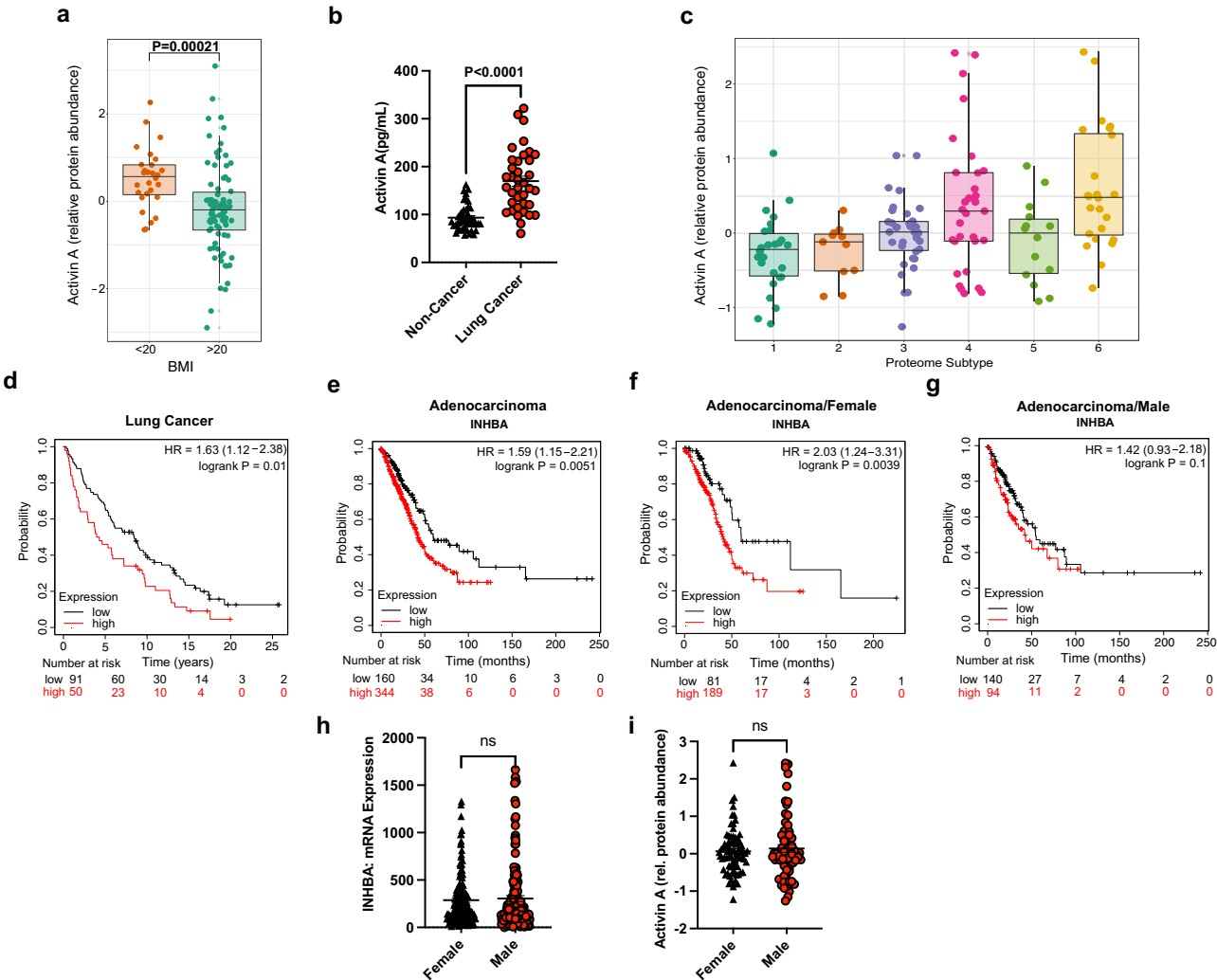

**Fig. 6 | Activin A expression in human lung adenocarcinoma is associated with poor prognosis only in female patients. a** Activin A relative protein abundance in lung tumors from CPTAC categorized by a BMI cutoff of 20 (BMI ≤ 20, n = 28; BMI > 20, n = 81). **b** Serum Activin A levels measured by ELISA in a cohort of lung cancer patients from Weill Cornell (n = 82). **c** Lung cancer Activin A relative protein abundance categorized by proteogenomic subtype defined by Lehtiö et al. (n = 141). **d** Kaplan-Meier (KM) plot with the probability of survival of patients (n = 141) with high or low Activin A protein levels published by Lehtiö et al. **e** KM plot with the probability of survival of patients with adenocarcinoma with high or low Activin A protein levels from TCGA (n = 504). **f, g** KM plots with the probability of survival of patients with adenocarcinoma with high or low INHBA expression levels divided in Females (**f**, n = 270) and Males (**g**, n = 234).

**h** TCGA Lung cancer mRNA expression of INHBA categorized by gender (n = 504). **i** Lung Cancer protein level of Activin A categorized by gender from Lehtiö et al. (n = 141). **h** For boxplots in **a**, **c**, horizontal bars within boxes represent medians. Tops and bottoms of boxes represent 25th and 75th per-centiles, and vertical lines extend to the 1.5× interquartile range. Comparison in **a**, **b**, **h**, **i** were performed with a two-tailed Student's t-test. Comparisons for the KM plots in **b**, **d–f** were done with the Log-rank Mantel-Cox test. Panel (**c**) was compared using One-Way ANOVA followed by Tukey's multiple comparisons test, p-values for significant proteome subtype comparisons: 4-1 = 0.004, 6-1 = 0.0001, 6-2 = 0.0085, 6-3 = 0.009, 6-5 = 0.023). Individual data points are independent biological replicates unless otherwise stated. Source data are provided as a Source Data file.

patients with lung cancer (Table 2) and found it to be significantly increased (Fig. 6b). Unfortunately, due to the small number of patients, it was not possible to draw any correlation with weight loss or prognosis.

A new classification of lung cancer was recently proposed by Lehtiö et al. based on tumor proteogenomic data[59]. Tumors from patients with lung cancer segregate into 6 molecular subtypes associated with specific immune-evasion mechanisms and therapeutic vulnerabilities. We found that Activin A was significantly high only in subtypes 4 and 6, two of the three subtypes with the worst survival (Fig. 6c). Interestingly, these subtypes are enriched in *STK11/LKB1* mutations (subtype 4) and squamous cell carcinomas (subtype 6), which are both features of the KL model[21]. In the Lehtiö et al. data set, patients with high tumor levels of Activin A had worse survival (Fig. 6d), in agreement with data from the TCGA (Fig. 6e)[60].

We found that blocking Activin A is only effective in female KL mice. Therefore, we searched the existing clinical databases for any evidence of sex-dependent effects of Activin A. Because Lehtio et al. is a relatively small sample, we used the TCGA data set. The TCGA allows segregation of the population by sex, and doing so reveals that only female patients have worse survival when Activin A gene expression is high in the tumor (Fig. 6f, g). Activin A expression in the tumor showed no sex difference at the mRNA (TCGA) or protein levels (Lehtio et al.), in agreement with the mouse model (Fig S7A). These data suggest that tumor-derived Activin A is associated with low BMI and poor prognosis in a sex-dependent manner in patients with lung cancer.

## Discussion
In this study, we performed a comprehensive analysis of the changes in food intake, peripheral organ metabolism, and TEE that occurs in mice

**Table 2 | Clinical demographics and tumor characteristics**

| Features | Total (N = 37) |
|---|---|
| Median Age (Range) | 73 (53–87) |
| Female Sex (%) | 59 |
| **Clinical Stage** | |
| IB | 6 |
| IIA | 11 |
| IIB | 7 |
| IIIA | 13 |
| **Pathologic Stage** | |
| **T** | |
| T1A | 3 |
| T1B | 2 |
| T2A | 16 |
| T2B | 5 |
| T3 | 10 |
| T4 | 1 |
| **N** | |
| N0 | 15 |
| N1 | 12 |
| N2 | 10 |
| **M** | |
| M0 | 32 |
| MX | 5 |
| **Histological Subtype** | |
| Adenocarcinoma | 26 |
| Adenosquamous | 2 |
| Atypical Carcinoid Tumor | 1 |
| Squamous | 6 |
| Other | 2 |
| **Differentiation** | |
| Well | 1 |
| Moderate | 16 |
| Poor | 16 |
| Undifferentiated | 1 |
| Other | 3 |

Age, gender, clinical stage, pathologic stage, histological subtype and grade of differentiation of a cohort of patients with lung cancer from Weill Cornell.

following induction of lung cancer. We show that the weight loss in KL mice with CACS is associated with anorexia and suppressed TEE. The reduction in TEE is more than what is predicted by changes in lean mass alone. This physiologic adaptation to weight loss has been observed in other mouse models of CACS and humans undergoing CR[61,62]. In this setting, the TEE reduction is due, in part, to reduced skeletal muscle glycolytic metabolism and improved work efficiency that can be prevented with low-dose leptin treatment[63,64]. Similarly, KL mice with CACS have atrophy of glycolytic muscle fibers[20], reduced muscle PFK activity, and low levels of leptin; these changes may improve muscle efficiency and contribute to the decrease in TEE during CACS.

Low levels of leptin are also known to contribute to the dramatic increase in spontaneous activity that we observed in mice following CR. This phenomenon, referred to as "food-seeking behavior" or "semi-starvation–induced hyperactivity," can be suppressed by replacing leptin or increasing housing temperature[65,66]. In mice with CACS, there is no hyperactivity despite the presence of anorexia and hypolepti-nemia. Furthermore, increasing the housing temperature para-doxically improves spontaneous activity in mice with CACS. These data

suggest that the normal neurohormonal pathways that regulate ther-mogenesis and spontaneous activity in mice are dysregulated.

Our data highlights the dramatic changes that occur to adipose tissue during CACS. KL mice develop increased rates of lipolysis, browning, and atrophy of the WAT adipocytes. Data from other mouse models and human studies of CACS show that WAT lipolysis is an essential feature of CACS[17,67,68], however, the role of browning is more controversial. In certain models, browning exacerbates the negative energy state[7,31], but this finding is not consistent with clinical studies in subjects with lung cancer nor our data from the KL mice[69–75]. We speculate that the browning observed in KL mice with CACS occurs in response to a reduction in core temperature mediated by BAT dys-function. BAT dysfunction has been observed in mouse models of diet-induced obesity where it can be reversed with fenofibrate, a PPARα-agonist[76,77]. Interestingly, we previously showed that fenofibrate can prevent CACS in KL mice so the role of BAT in this response needs further study[20,76,77].

Our results identify the distinct alterations in skeletal muscle metabolism that occur during weight loss from CACS in comparison to weight loss from CR. In both conditions, we see similar reductions in body weight, skeletal muscle mass, and TEE; however, the reduction in exercise tolerance and markers of oxidative metabolism only occur in CACS. It is unclear if the lack of spontaneous activity and limited exercise tolerance is due to malaise (*i.e.*, centrally-mediated), reduced cardiac output, or a primary deficit in skeletal muscle. In support of the latter, Kamei et al. have shown that overexpression of Forkhead box protein O1 (FoxO1) in skeletal muscle is enough to suppress sponta-neous activity, and we have previously shown that the muscles from KL mice with CACS have increased expression of this gene. We also found evidence for a reduction in the gene expression and protein abun-dance of several proteins involved in the electron transport chain in the EDL of mice with CACS; however, there was no change in the oxygen flux of the permeabilized EDL when measured ex vivo. Given that the EDL contains significant numbers of non-atrophied type IIA fibers[20], we speculate that the activity of these highly oxidative fibers is masking any change in ex vivo oxygen flux.

The changes to skeletal muscle and adipose tissue metabolism were associated with high levels of Activin A[50,51]. We identified increased tumor expression of *Inhba* and activation of the TGF-β/SMAD pathway in the skeletal muscle using unbiased transcriptomics. Activin A has been previously established as a pro-cachectic factor, and ActRIIB-Fc treatment is known to improve muscle mass and survival in cachectic mice bearing tumor allografts[78–82]. In KL mice, ActRIIB-Fc increased lean mass and spontaneous activity, but had no benefit on fat mass or survival unless it was given with an agent that effectively improved appetite (i.e., Ana). Ana therapy in isolation trended towards improved food intake and reversal of weight loss with no significant increase in survival. Similarly, RCTs in humans showed significant improvement in body weight and lean mass but no changes in overall survival. Considering the results of our study, we surmise that improved survival in humans could be achieved if Ana would be combined with agents that block cachectic factors like ActRIIB-Fc.

The mediator(s) of anorexia in the cachectic KL mice remain unknown and are an area of active investigation. In this study, we show that anorexia is not responsive to anti-GDF-15 therapy despite increased levels in the serum. This lack of response is likely because the concentrations we measured are relatively low compared to other models where anti-GDF15 therapy has shown benefit[67,83,84]. Whatever the etiology, we are encouraged by the ability of Ana to override the anorexia program and look forward to future clinical interventions where Ana is paired with anti-cachectic therapies.

The beneficial effects of Ana/ActRIIB-Fc were limited to female mice. Sex discrepancies have been observed in other models of muscle wasting such as disuse atrophy and cardiac cachexia[85–87]. Still, this finding was surprising as Activin A is upregulated to the same degree in

both male and female mice. Similarly, the expression of Activin A in human tumors is the same for both sexes yet, its impact on survival is only apparent in females. These data suggest that tumor-secreted Activin A is having sex-dependent effects on the host that lead to CACS. There are known differences in body composition, EE, and peripheral organ metabolism between men and women driven, in part, by sex hormones[88]. While peripheral estrogen receptor blockade with fulvestrant did not abrogate the response to the Ana-ActRIIB-Fc therapy, the surgical excision of the ovaries did. Activin A is known to stimulate the release of follicle stimulating hormone from the pituitary and control various ovarian functions, including follicular growth and development and ovarian steroidogenesis that could be playing a role in the sex dependent response to the combination therapy[89]. Additional studies are required to identify the specific role of the ovary in the response to the combination therapy.

Human CACS is a heterogenous condition with patients experiencing disparate degrees of anorexia, inflammation, and muscle wasting[18,90]. It is unclear which pre-clinical models best reflect each clinical subset. Here, we use existing clinical oncology databases to identify lung cancer subtypes that are best modeled by the KL mice. These subtypes (4 and 6 from Lehtio et al.[59]) have high levels of Activin A and poor overall survival. Future clinical studies could use the molecular features of these subtypes and blood Activin A levels as enrollment criteria. For example, there are late-stage clinical compounds available for the immediate translation of our findings. Ana was recently approved by the pharmaceutical regulatory authority of Japan for the treatment of patients with CACS, and bimagrumab, a fully human monoclonal antibody that prevents ligand binding to ActRIIB, is safe and increases lean mass in adults with sarcopenia and metabolic disease[53,91,92]. Our data suggests that the combination of Ana and bimagrumab would be effective in select patient populations.

Our study has several limitations. First, our tissue analyses were limited to samples obtained from the studies' endpoint, a time where numerous metabolic pathways have been perturbed. Therefore, we are unable to distinguish the primary, causal events from the secondary response. Second, our intervention trials enrolled mice that lost 15% body weight. Our prior work suggests that this threshold marks the appearance of systemic metabolic dysfunction, and our goal was to treat existing CACS as opposed to preventing mice from developing CACS. Nevertheless, this amount of weight loss may be considered "refractory" in other models.

The degree of survival improvement we observed with the Ana/ActRIIB-Fc therapy is on-par with the effects of chemotherapy and immunotherapy in this model yet, without impacting tumor growth[93,94]. These data highlight the power of multimodal therapy targeting both anorexia and peripheral organ dysfunction when they are both present. Future clinical studies could address the feasibility of this approach.

## Methods

### Study approval
All animal studies were approved by the Institutional Animal Care and Use Committee (IACUC) of Weill Cornell Medical College and maintained as approved by the Institutional Animal Care and Use Committee (IACUC) at Weill Cornell Medicine (NY) under protocol number 2013-0116.

### Experimental model
Kras$^{G12D/+}$; Lkb1$^{f/f}$ mice have been previously described[21] and were further backcrossed to FVB mice. Mice were housed in a 12 h light/dark cycle at 22 °C or 30 °C ambient temperature and received rodent chow (PicoLab Rodent 20 5053; Lab Diet, 3.43 kcal/g) and free access to drinking water. Tumors were induced in adult (12- to 20 week-old) FVB mice via intranasal administration of 75 μL of PBS containing $2.5 \times 10^7$ pfu of Adenovirus CMV-Cre (Ad5CMV-Cre) obtained from the

University of Iowa Gene Transfer Vector Core (Iowa City, IA) and 1 mM CaCl$_2$. Mice are defined as +CACS if they lost more than 15% of body weight from their peak weight over the course of the cohort. If they do not reach this cutoff, they are classified as -CACS.

### Tissue collection.
Whole blood glucose was measured using a point of care glucose meter and blood from the tail vein before CO$_2$ asphyxiation. Following euthanasia, whole blood was collected via cardiac puncture and placed into pre-treated tubes for serum/plasma isolation. Next, the liver, gonadal adipose, kidney, and skeletal muscles (gastrocnemius, quadriceps, tibialis anterior, EDL, and Soleus) were dissected, weighed, and flash-frozen in liquid nitrogen or fixed in 10% formalin. The frozen tissues were subsequently stored at −80 °C until further processing.

### Metabolic cage analyses.
Mice were individually housed at 22 or 30 °C and subjected to indirect calorimetry for a period of 3 consecutive days under a 12 h light-dark cycle using the Promethion Metabolic Cage system (Sable Systems, USA). During this period, food and water intake, spontaneous activity, and volume of oxygen consumed and carbon dioxide produced were measured. This data allows us to calculate total energy expenditure (TEE) and the respiratory exchange ratio (RER). To compare TEE between WT, -CACS and +CACS in Fig. 1g, we normalized TEE to lean mass using ANCOVA following the recommendations of the National Mouse Metabolic Phenotyping Center (MMPC, www.mmpc.org) and others[36,37]. We recorded the KL mice in two phases: 1- Acclimation, first 24 h of measurement; 2- Fed, 24 h after the end of phase 1, mice were fed ad libitum. A penalized spline regression model was used to estimate the resting energy expenditure (REE), activity energy expenditure (AEE), and caloric cost of activity (CCA), as previously described[46]. To calculate CCA, we fit a simple linear regression model between activity rate and total energy rate for each mouse. The slope of each line is the CCA for each respective mouse. To calculate AEE, the mouse's activity rate was multiplied by[37] its CCA. REE was calculated by subtracting AEE from TEE. REE and AEE were then smoothed using a second-order polynomial smoothing spline. This method allows for the calculation of AEE while taking into account time-varying REE.

### Food intake.
When mice were housed in the metabolic chambers, real time food intake was assessed using high precision sensors that detected changes in the crushed food pellet mass. The system has a built-in 'crumb catcher' tray that eliminates spilling and reduces caching behavior, providing more accurate and repeatable results. When not in the metabolic chambers, mice were single housed starting 2 weeks from tumor induction and the mass of the whole food pellets was recorded 3 times per week approximately between 10 and 12 am when food pellets were also changed. The residual food crumbs at the bottom of the cage was collected and weighed by filtering out the bedding using a baking sifter; however this amount was never significant.

### Body composition analysis.
Mice were weighed and body composition (fat mass, free fat mass, and water mass) was measured using an EchoMRI-100H 2n1 with a horizontal probe configuration (EchoMRI, Houston, TX).

### Exercise capacity test.
Mice were acclimated (30 min at 8 m/min) to a motorized treadmill one week before the maximal exercise capacity test. Shocking grids with frequency set at: 75 per minute and intensity at: 45% (3.4 mA) were located at end of the treadmill to force the mice to run at their maximum. On the day of the test, the protocol was initiated with 3 min acclimation without any speed. Start speed was set to 8 m/min followed by incremental adjustments of 2.5 m/min every 3 min until fatigue was reached. Fatigue was defined as the mouse

being stationary on the shocking grid for 20 s with no attempts to climb off the treadmill. Maximum speed, time and laps were then recorded and used to calculate, total time, total distance, and work. Lactate was quantified before and after the exercise protocol using a point of care device (Nova Biomedical).

**Therapeutic trials.** The GDF15 mAb and ActRIIB-Fc were provided by Pfizer (Boston, USA) and have been previously described[95,96]. A pilot experiment was performed to assess safety in 24 WT mice. The mice were randomly assigned into 3 groups: IgG control (20 mg/kg) weekly (QW) via subcutaneous (SQ) injection, a weekly injection of GDF15 mAb (10 mg/kg, QW, SQ) alone, or a combination of GDF15 mAb and ActRIIB-Fc (20 mg/kg, QW, SQ), for 2 weeks. Anamorelin fumarate (Ana) was obtained from BOC Sciences (New York, USA) and used at 30 mg/kg as previously described[62,97,98]. The combination of GDF15 mAb and ActRIIB-Fc increased food intake, body weight, and lean mass with no effects on TEE, RER, or cumulative activity. There was no clear toxicity observed. Next, we performed 2 prospective, randomized, controlled, intervention trials using KL tumor-bearing mice (RCT 1 and RCT2). RCT1 was performed using the GDF15 mAb alone or in combination with ActRIIB-Fc. RCT2 was performed using Ana alone or in combination with ActRIIB-Fc. Based on prior data, we estimated that the RCTs would require 7 mice per group to detect a 20% change in mean weight loss ($\alpha = 0.05$, $\beta = 0.90$). Following induction, if mice lost 10−15% of their peak weight, they were randomized to a treatment arm. The randomization was performed in blocks of 6 and stratified by sex[99]. The pre-specified primary outcome was the percent weight loss at 2 weeks following the start of treatment (End). If a mouse met endpoint criteria (loss of 30% body weight or body composition score < 2), before the End date, then −30% would be used for their End value. Secondary outcomes included food intake, overall survival, body composition, food intake, spontaneous activity, skeletal muscle mass, and white adipose tissue mass. Overall survival was calculated from the date of induction. In RCT1, 39 mice (22 males, 17 females) were induced, and 29 mice (17 males, 12 females) underwent randomization. In RCT2, 50 mice (26 males, 24 females) were induced, and 46 mice (23 males, 23 females) underwent randomization. Given the heterogenous timing of weight loss and death at the individual level, not all mice were available for each secondary endpoint. The sample size for each secondary measure is included in the legends.

**Bilateral ovariectomy.** 12−20 week old female mice were randomized to sham surgery ($n = 20$) or ovariectomy ($n = 21$) 4 weeks before tumor induction with $2.5 \times 10^7$ pfu of Adenovirus CMV-Cre. A parasagittal incision was made in the abdominal musculature ~1 cm lateral to the spine in order to access the peritoneal cavity. The ovary and oviduct were exteriorized and the oviduct was ligated before removing the ovary with scissors. The remaining tissue and uterine horn are replaced in the peritoneal cavity. The opposite side is accessed through the same skin incision and the other ovary is then removed in a similar manner. The skin is apposed using sterile wound clips in a subcuticular or intradermal pattern. Mice were examined at least once daily for the next 72 hs post-procedure for any symptoms of pain or illness and meloxicam (2 mg/kg every 24 h SC) was administered for analgesia. Exactly the same procedure was done for the sham surgery but the ovaries were left intact and replaced in the peritoneal cavity.

**Caloric restriction experiments.** WT (non-tumor bearing) male mice were single housed and randomized either to the control group with ad-libitum access to food pellets (PicoLab Rodent Diet 20 5053) or to the caloric restriction group were mice were fed at 6.00 pm with 2.33 grs of food pellets. This amount of food is equivalent to 8 kcal/day (3.43 Kcal/gm), that is the average caloric intake of cachectic mice as shown in Fig. 1h. Total body weight was measured every other day and

when mice reached 25% weight loss, body composition was performed by echoMRI followed by indirect calorimetry in the Promethion Metabolic Cage system as described above. Mice that developed a body composition score of 2 or less were re-fed and excluded from the study.

**Humane endpoints.** Our study was performed with the approval of the Institutional Animal Care and Use Committee, which agreed to 30% weight loss as a humane endpoint. In a prior study[20], we characterized the metabolic changes that occur in KL mice with and without CACS. We found that systemic metabolic dysfunction, anorexia, and muscle atrophy only occurs after 15% weight loss. Therefore, this threshold is used as the defining criteria for CACS in this model. As shown in Fig. 1a, the mice reach 30% weight loss 7-14 days after the onset of CACS. This period was determined to be an adequate duration for therapeutic intervention. The tumor bearing mice in our trials are monitored weekly for weight, food intake, and body composition score (a score <2 is another humane endpoint). When mice reach 20%, they are weighed and monitored daily for poor body composition. In some instances, mice rapidly lose weight and exceed the 30% weight loss criteria at the next measurement. Those mice are euthanized immediately.

**Serum and tissue metabolites.** Blood was centrifuged (10,000 × g for 10 min at 4 °C), and the serum or plasma was stored at −20 °C. Serum β-hydroxybutyrate, TG (Stanbio Laboratory), and NEFA (Wako Life Sciences) were determined using commercially available kits. Serum insulin, corticosterone (APLCO Diagnostics), Leptin (Milipore, cat.# EZML-82K), Activin-A (DAC00B, R&D Systems), GDF-15 (MGD150, R&D Systems) levels were quantified by ELISA. Plasma epinephrine and norepinephrine were measured by HPLC via chromatography data station by the Vanderbilt Hormone and Analytical Services Core (sensitivity of 0.5 ng/ml for the mouse samples)[100,101]. Serum T3 and T4 were determined by radioimmunoassay in a double antibody technique. T4 was measured by using I125-labeled T4 (MP Biomedicals Cat# 06B257231) and 1st antibody developed in rabbit (Sigma Cat# T2652). T3 was measured by using I125-labeled T3 (MP Biomedicals Cat# 06B254282) and 1st antibody developed in rabbit Sigma Cat# T2777). Tissue metabolites (including T3 and T4) were extracted from gastrocnemius (whole muscle) using 80% methanol[102]. Targeted LC/MS analyses were performed on a Q Exactive Orbitrap mass spectrometer (Thermo Scientific) coupled to a Vanquish UPLC system (Thermo Scientific) as previously described[20]. Metabolites were identified on the basis of exact mass within 5 ppm and standard retention times. Relative metabolite quantitation was performed based on the peak area for each metabolite. All data analyses were done using scripts written in-house by the WCM Proteomics and Metabolomics Core Facility.

**RNA sequencing and analysis.** Total RNA was extracted from gastrocnemius (whole muscle) and gonadal WAT (whole depot) using TRIzol (Thermo Fisher), followed up by a clean-up step using RNeasy kit (Qiagen). One microgram of total RNA of each sample was submitted to the WCM Genomics Resources Core Facility. Raw sequenced reads were aligned to the mouse reference GRCm38 using STAR (v2.4.1d, 2-pass mode) aligner and Raw counts were obtained using HTSeq (v0.6.1)[103,104]. Differential expression analysis, batch correction and principal component analysis (PCA) were performed using R Studio Version 3.6.3 and DESeq2 (v.1.30.1). Gene set enrichment analysis (GSEA) analysis was performed using GSEA (v4.1.0) JAVA based application. Volcano plots were done using EnhancedVolcano (v1.11.5). The percentage of browning in the WAT was assessed using the online webtool BATLAS, an algorithm capable of estimating the fraction of brown adipocyte content per sample based on RNA-Seq[32]. The list of genes associated with unfavorable prognosis in the TCGA data set was obtained from the ProteinAtlas (https://www.proteinatlas.org/).

## Table 3 | Primer Sequences

| Primer | Sequence 5'-3' |
|---|---|
| m-PGC1α-F | AGCCGTGACCACTGACAACGAG |
| m-PGC1α-R | GCTGCATGGTTCTGAGTGCTAAG |
| m-Rer1 –F | GCCTTGGGAATTTACCACCT |
| m-Rer1 –R | CTTCGAATGAAGGGACGAAA |
| m-Rpl41 –F | GCCATGAGAGCGAAGTGG |
| m-Rpl41 –R | CTCCTGCAGGCGTCGTAG |

Sequences (5'-3') of the primers used in SYBR green qPCRs.

**RT-qPCR.** Total RNA was extracted from total muscle EDL and Soleus using the method described above and cDNA was synthesized using SuperScript VILO Master Mix. cDNA was amplified using the Applied Biosystems TaqMan Gene Expression Assays (Thermo Fisher) with primers for the following genes: Ucp1; Lep; Rer1 and Actb. cDNA was amplified using the Applied Biosystems SYBR™ Select Master Mix (Thermo Fisher) with the primers detailed in Table 3. Relative mRNA expression levels (2^-ΔCt) were normalized to mean of the -CACS group.

**Mitochondrial DNA.** Total DNA was isolated from EDL and Soleus using Qiagen Dneasy Blood and Tissue Kit and treated with RNase A according to the manufacturer's instructions. The mitochondrial DNA content (*mt-Nd2, NADH dehydrogenase 2, mitochondrial*) relative to nuclear DNA (*Pecam1*, platelet/endothelial cell adhesion molecule 1) was determined by quantitative real-time PCR using the Applied Biosystems SYBR™ Select Master Mix (Thermo Fisher). Relative mtDNA content was determined using the ΔΔCt method.

**Histology.** BAT, gWAT and quadriceps were fixed with 4% paraformaldehyde solution in PBS followed by storage in 70% ethanol and embedding into paraffin. Four-micrometer sections were cut for staining with H&E. Samples were also stained for UCP1 through immunohistochemistry using 1:200 dilution of a recombinant anti-UCP1 antibody [EPR20381] (ab209483) that we validated against UCP1 knockout tissue. UCP1 staining and fat quantification in the BAT were perfomed using QuPath 0.2.3. Adipocyte size in the WAT was calculated using the ImageJ plugin Adiposoft. Muscle fiber area was measured using Adobe Photoshop.

**Western blots and antibodies.** EDL and Soleus (whole muscle) were lysed using lysis buffer containing 50 mM Tris·HCl (pH 7.4), 150 mM NaCl, 1 mM EDTA, 10% glycerol, 1% Nonidet P-40, 0.5% Triton X-100, and 1 tablet (per 10 mL) of protease and phosphatase inhibitor. Protein extracts (50 μg) were separated by 4–12% NuPAGE Bis-Tris or 4–20% NuPAGE Tris-Glycine gels and transferred to 0.45 μm PVDF membranes with wet transfer cells (Bio-Rad Laboratories). After 1 h of blocking with Tris-buffered saline with 0.1%(vol/vol) Tween 20 containing 3%(wt/vol) BSA (TBST), membranes were incubated overnight at 4 °C with antibodies against UCP1 (ab209483); PGC1 alpha (ab54481); LDHA (CST #2012); Phospho-LDHA (Tyr10) (CST #8176); Pyruvate Dehydrogenase (CST #3205); Anti-Pdhe1α (Ser293) (ab92696); Total OXPHOS Cocktail (ab110413); GAPDH (Proteintech 10494-1-AP), VDAC (CST #4661); and α-Tubulin (DM1A) (CST #3873) at a 1:1,000 dilution in 3% BSA followed by a TBST wash and the appropriate secondary antibody (1:10,000) for 1 h at room temperature. The signals were detected on HyBlot CL Autoradiography Film (Denville Scientific) with SuperSignal Western Blot enhancer solution (Thermo Fisher), scanned at 600 dpi resolution, cropped with Adobe Illustrator 2020 (Adobe). Uncropped scans of the Western blots are provided in the Source Data file.

**Assessment of oxidative capacity in permeabilized muscle fibers.** Oxidative phosphorylation (OXPHOS) and electron transport (ET) capacity were determined ex vivo from permeabilized mixed gastrocnemius fibers as described previously[105]. Briefly, intact soleus and EDL muscle was collected and immediately placed into BIOPS (50 mM K + -MES, 20 mM taurine, 0.5 mM dithiothreitol, 6.56 mM MgCl₂, 5.77 mM ATP, 15 mM phosphocreatine, 20 mM imidazole, pH 7.1, adjusted with 5 N KOH at 0 °C, 10 mM Ca−EGTA buffer, 2.77 mM CaK₂EGTA + 7.23 mM K₂EGTA; 0.1 mM free calcium) solution on ice. The muscle bundles were then mechanically separated under a dissection microscope, placed into fresh BIOPS containing saponin (5 mg/mL), and gently agitated at 4 °C for 20 min. The fibers were then transferred to a mitochondrial respiration medium (MiR05; 110 mM sucrose, 60 mM K + -lactobionate, 0.5 mM EGTA, 3 mM MgCl₂, 20 mM taurine, 10 mM KH₂PO₄, 20 mM HEPES adjusted to pH 7.1 with KOH at 37 °C; and 1 g/l de-fatted BSA), blotted on filter paper, and weighed. 2−5 mg of permeabilized fiber bundles were transferred into the oxygraph chamber containing 2 mL of MiR05 until background respiration was stable. OXPHOS and ET capacity were measured using the following concentrations of substrates, uncouplers, and inhibitors: malate (2 mM), pyruvate (2.5 mM), ADP (2.5 mM), glutamate (10 mM), succinate (10 mM), tetramethyl-p-phenylenediamine (TMPD, 0.5 μM), ascorbate (2 mM), carbonylcyanide-p-trifluoromethoxyphenylhydrazone (FCCP, 0.5 μM increment), rotenone (75 nM), antimycin A (125 nM) and sodium azide (200 mM).

**Lipolysis assay.** The entire gWAT fat depot was isolated, weighed, and cut. 30−50 mg of tissue was incubated in 600 ul per well of filtered lipolysis medium (DMEM, 2% BSA) with or without Isoproterenol (1 μM). At various time points during incubation at 37 °C, the medium was collected and glycerol concentration was measured using a free glycerol determination kit (Sigma-Aldrich) according to the manufacturer's instructions. Sample absorbance was measured at 540 nm, using the Epoch™ 2 Microplate Spectrophotometer (BioTek), and glycerol content was normalized to the initial tissue weight.

**Muscle enzyme activity.** Citrate Synthase (CS) and 6-phosphofructokinase (6-PFK) enzyme activity were quantified from EDL and soleus protein lysate using commercially available colorimetric assay kits (CS0720 Sigma-Aldrich Citrate Synthase; and 6-Phosphofructokinase Activity Assay - ab155898) according to the manufacturer's instructions and quantified using an Epoch™ 2 Microplate Spectrophotometer (BioTek).

**Statistics and reproducibility.** Data are expressed as mean ± standard error of the mean (SEM). Statistical significance for normally distributed data was determined using Student's t-tests for comparisons of 2 groups or analysis of variance (ANOVA) followed by Fisher LSD post-hoc tests for comparisons of 3 or more groups. For metabolic cage analyses, ANOVA followed by Tukey's multiple comparisons test were used unless otherwise stated in the text. Significance was set at $P < 0.05$. All western blots in the manuscript contain at least 4 samples per condition to ensure that representative results are shown. Statistical analyses were performed with Prism 7 (GraphPad Software), R (v.4.0.5) and python (v.3.9.1). Quantification of Western blots was performed using ImageJ 1.53a. Kaplan Meier survival plots were made using https://kmplot.com/ and analyzed using Log-rank Mantel-Cox test.

### Reporting summary
Further information on research design is available in the Nature Research Reporting Summary linked to this article.

## Data availability
All data to understand and assess the conclusion of this research are available in the main text, supplementary materials, Source Data file or GEO Databases. Previously published gastrocnemius RNA-Seq files used in this manuscript are available in the GEO Database(GSE107470)[20]. The newly generated RNA-Seq files used here can also be accessed in the

GEO Database (accession number GSE165856). All available reagents are available from the Lead Contact under a material transfer agreement with Weill Cornell Medicine. Source data are provided with this paper.

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

## Acknowledgements

This work was supported by a grant from the Lung Cancer Research Foundation (M.D.G.), NIH K08 CA230318 (M.D.G.), the 2020 AACR-AstraZeneca Lung Cancer Research Fellowship Grant Number 20-40-12-DANT (E.D.), NIH R35 CA197588 (L.C.C.), and institutional support from Weill Cornell Medicine. We thank the Metabolic Phenotyping Center, the WCM Proteomics and Metabolomics Core Facility, the Electron Microscopy and Histology Core Facility, the Vanderbilt Mouse Metabolic Phenotyping Center (NIH DK059637) and Diabetes Research and Training Center (NIH DK020593), the Louisiana Clinical and Translational Science Center (U54 GM104940) and the Scott Rodeo for use of the rodent treadmill. The GDF-15 antibody and ActRIIB-Fc were provided by Pfizer, Inc. Catecholamines assays were performed by the VUMC Hormone Assay and Analytical Services Core which is supported by NIH grants DK059637 and DK020593.

## Author contributions

A.L.Q., E.D., S.R., E.R.M.Z., R.J.L., J.M., A.M., M.A., C.D.H., C.J.B., C.L.A., and M.D.G. performed experiments. A.L.Q., E.D., S.R., E.R.M.Z., R.J.L., J.M., C.D.H., C.L.A., Z.W., D.E.C., L.C.C., and M.D.G. designed the project. A.L.Q., E.D., E.R.M.Z., J.M., C.J.B., C.D.H., J.P.K, A.M., M.A., C.L.A., G.G., and M.D.G analyzed the data. A.L.Q, E.D., and M.D.G. wrote the paper. All authors read, edited, and approved the manuscript.

## Competing interests

L.C.C. is a founder, shareholder, and member of the scientific advisory board of Agios Pharmaceuticals and a founder and former member of the scientific advisory board of Ravenna Pharmaceuticals (previously Petra Pharmaceuticals). These companies are developing novel therapies for cancer. L.C.C. has received research funding from Ravenna Pharmaceuticals. M.D.G. reports personal fees from Novartis, Petra Pharmaceuticals, and Bayer. He has received research support from Pfizer Inc. L.C.C. and M.D.G. are inventors on patents unrelated to the scope of the current work. L.C.C. and M.D.G. are co-founders and shareholders in Faeth Therapeutics. Z.W. is a full-time employee of Pfizer Inc. All other authors report no competing interests.
