## [Peer Review File · Nature Communications]

Blocking ActRIIB signaling and restoring appetite reverses cachexia and improves survival in mice with lung cancerReviewers' comments:

Reviewer #1 (Remarks to the Author):

Summary

The manuscript by Lima Queiroz et al. explores the contribution of anorexia and energy expenditure to cancer-associated cachexia in a mouse model of lung cancer. The authors demonstrate that lung-cancer induced cachexia was characterized by extensive remodeling of white adipose tissue (WAT) with elevated levels of UCP1, reduced leptin levels and elevated lipolysis. However, systemic oxygen consumption and energy expenditure was found to be lower as compared to non-cachectic counterparts. Based on high throughput sequencing, skeletal muscle was characterized by inhibition of mitochondrial function, correlating with reduced exercise capacity. As compared to cancer cachexia, simply caloric restriction (CR) triggered distinct changes in WAT and skeletal muscle metabolism but displayed similar effects on systemic energy expenditure and food intake. In an attempt to counteract/prevent cancer-induced cachexia, the authors then used a combination treatment using anti-GDF15 antibodies and inhibition of Activin A, two well established mediators of cancer-induced cachexia. However, treatment of cachectic lung cancer mice with this combination did not lead to major metabolic phenotypes or improvements. Finally, combination of Activin A inhibition with the ghrelin-agonist anamorelin fumarate (Ana) resulted in increased food intake and protected against the loss of fat and lean mass. Of note, this combination therapy was particularly effective in female animals, also improving overall survival. The authors conclude that effective cachexia therapy requires gender-specific, multimodal therapies.

General Comments

Cancer-induced cachexia clearly represents a major unmet clinical need with currently no pharmacological therapy approved. In this regard, the study by Lima Queiroz et al. addresses an important and timely topic. Overall, the study employs state-of-the-art technology, appropriate models and is clearly structured and well written. On the other hand, a number of major issues require additional attention by the authors: a) The reported reduction in systemic energy expenditure is interesting as it contradicts other studies in the field. However, these findings have been reported previously (Rohm et al. Nat Med. 2016), and thus lack a certain degree of novelty. b) The manuscript in general remains at a rather descriptive level. In particular, the mechanisms of gender-specific responses to pharmacological interventions are interesting and should be explored in more mechanistic details. c) The relevance of the reported findings for the human pathology remain unclear. Do WAT and skeletal muscle from lung cancer patients display similar features as observed in this animal model? Overall, despite the interesting research question, the manuscript in its current version reflects a rather preliminary study, lacking mechanistic details (gender-specific aspects), and the main conclusions are common sense in the field already. Additional data on the listed major concerns will strengthen the case for publication in the future.

Specific Comments

1. Please explain the rationale for using exactly these pharmacological combination therapies (amongst all other possible combinations, and beyond their simple availability). The use of the combination therapies is not deducted from any specific data obtained by the authors, and thus it remains unclear why this road was taken.

Reviewer #2 (Remarks to the Author):

Overall, the study by Queiroz et al is comprehensive, addresses an important biological problem, and relevant therapeutics, and provides some novel insights into cachexia pathophysiology. There are several issues that should be addressed:

According to the AAALAC guidelines for welfare and use of animals in cancer research, animals should not be allowed to exceed 20% weight loss without euthanasia. In this study, it appears that

many animals went beyond this, with some losing nearly 40% of body weight (figure S1a, figure 5b). Please include a statement about the exception to the AAALAC guidelines, and how humane endpoints were monitored in the study. It is worth discussing the implications of such severe weight loss in the experimental animals- this is well beyond any cachexia definition in humans and is an extreme physiological state. Indeed, in the drug trials, the animals had already passed 15% weight loss before initiation of therapy. Is it possible this represents “refractory cachexia” given the severe weight loss in this cohort? Finally, were the tumor masses in these animals recorded?

Although a great deal of the data in this paper is dependent on accurate measurement of food intake, no description of the methods for determining food intake are provided. Were animals individually housed? Was food intake automated? Solid food or powdered food? Was bedding screened for food waste? The latter is critical as it is evident from multiple prior studies that food wasting and shredding is a common behavior in sick mice.

In figure 1c,f the -CACS animals have much higher activity at 30C than at 20C, yet the energy expenditure at 30C is much lower than at 20C. Presumably this is because of a difference in BMR. It would be helpful to estimate the difference in BMR under these two conditions, and put that into the context of what is already known about the impact of ambient temperature on metabolic rate in mice. The RER data are also a bit puzzling- the food intake of the +CACS 22C and -CACS 30C are the same, yet their RER is quite different. How might this be explained?

The method of dividing mice into +CACS and -CACS is confusing. Presumably, this is done retrospectively, with animals reaching a 15% weight loss being the +CACS group. Review of figure 1A suggest that in fact all mice are losing weight, with the -CACS “late” group being at or below the +CACS “mid” group. Isn't this just a reflection of the stage of CACS development? That is, the rate at which the +CACS group is losing weight is just a bit faster than the -CACS group, perhaps due to differences in the timing, degree, and anatomical location of the tumors? What is the final tumor mass of these various groups? For RNA seq studies, what stage of weight loss were the +CACS animals relative to the -CACS? If end-stage, with a minimum of 15% weight loss, then the limitations of interpretation of these data should be discussed. Specifically, this would represent severe, refractory cachexia in a human, and observed changes in gene expression are more likely to represent that stage than to represent causal, driving pathways that would have been active much earlier in the cachexia process.

Lung cancers in humans, and previous studies with lung cancer models in mice show increased levels of PTHrP in circulation, and at least one study implicates this protein in the browning of WAT. If samples remain available, it is therefore reasonable to assay the levels of this endocrine molecule in these studies.

There were two females in the GDF15 study that apparently responded to treatment. Is anything known about their tumor burden? Did they have improvements in activity and food intake?

In S11 and S12, the numbers of animals in the treatment arms does not seem to match the number analyzed in the fat and lean mass (c, d in each figure). For example, there is only a single animal point in figure S12 c,d for the combined therapy. Why are the data for the other animals not shown? Similarly, in figure 5, the numbers in the data for ending weight do not match the numbers for the fat mass and lean mass and one of the dots for weight loss percent in the combined treatment group does not have a connecting line to a starting weight. The numbers in the lung masses (Fig 5e) also do not line up with the numbers in the lean and fat mass.

The study of combined Ana/ActRIIb was divided into sex post hoc. Was this study adequately powered to address this variable? In general, the dangers of post hoc analysis based on a single experiment should be addressed in the discussion.

The “loss of insulation” hypothesis raised in the third paragraph of the discussion is interesting- how is this consistent with the observations in pure calorie restriction? Does this imply that no browning

should be observed in animals in a thermoneutral environment?

For the forced treadmill studies, it is important to point out that mice are far more limited by cardiac output than humans are, and cardiac reserve and heart function may be more limiting in this study than skeletal muscle function. Any information on heart muscle would be useful to include, if available. Furthermore, cardiac function as a potential driver on mobility should be added to the discussion.

The statement is made that the BAT was smaller in the CACS mice, but that the size of the adipocytes did not change. How is this reconciled?

Gender is not applicable to mice- this should be changed to "sex" throughout. Although the issue of gender specificity of pathology and therapeutic response is interesting, what is being studied here is sexual dimorphism.

Under "metabolic cage analysis" the sentence implies that both O₂ and CO₂ are "consumed". Presumably you mean "produced" for the CO₂.

Need to confirm that only a single transcript was amplified in the qPCR studies, particularly because this is done with SYBR rather than a primer/probe pair.

It is currently unclear throughout the manuscript under what thermal conditions the various studies were performed. The materials and methods just states "at 22 or 30 C" but does not state when these two different conditions were used.

In figure 3A, the Y axis should read "percent baseline weight", not "% weight loss".
Figure S8A Y axis units should be mg not g.

Reviewer #3 (Remarks to the Author):

Comments to the author

The manuscript comprehensively examines the relationship of food intake, spontaneous activity, and energy expenditure on cancer-induced cachexia, body composition changes, and metabolic dysfunction in a preclinical lung cancer model. Further, they tested the effects of a GDF-15, ghrelin agonist, and ActRIIB-fc blocking of TGFB / activin. The analysis includes an unbiased screen of WAT and skeletal muscle. The investigative team and corresponding author are productive experts in examining diet and cancer. The clearly proposed but undervalued descriptive premise for investigating the regulation of energy expenditure with cancer cachexia is still strong and important.

Comments

Overall, the study is comprehensive, contains valuable information, and is very well conducted. Important information is abundant for the cachexia field. The primary issue with impact is that almost none of this novelty can be grasped from the introduction, which reads as a much more simplistic descriptive study. The nuances of sex, caloric restriction, successes, and failures of the treatments used, the novelty of the model, and the design that incorporates treatment are left for the reader to struggle with and discover. Only the most experienced cachexia researcher may grasp the impact. Strengths include the preclinical model, CLAMS analysis, and functional testing. Weaknesses include the mechanistic insight provided from the therapeutic nature of the treatment administration and how this contributes to the existing body of knowledge for these treatments. The fact the overall presentation resonates as model-specific serves to lower the potential impact. The results presentation is exceptionally complex. A more focused presentation with a more robust rationale within specific areas of emphasis will increase the study's impact on investigators in the field.

There is far too much supplemental data, which is not sufficiently explained, seemingly being a data dump so that it can be called “published.” This should be substantially reduced to the data that can be sufficiently explained in the manuscript and provide direct support.

The introduction carefully defines the metabolic terms and lung cancer model. Still, it is superficial in detailing the treatments and inhibitors described in the introduction's lengthy results portion. Still, the premise related to what is known and their function is insufficient.

The equivocal role of anorexia in several preclinical models of cachexia is not clearly defined. Furthermore, there is a current emphasis on the initiation and reversal of cachexia in the design, but not in the rationale. The mechanisms investigated early drivers cachexia initiation or limited to refractory cachexia, which is not clearly explained. The results in figure 1 are confusing related to early and late. Please clarify the presentation of repeated measures in the same mice, which seems to be a strength of the design.

The possibility that cachexia can be initiated and metabolic dysfunction occurs before many classic signs of overt refractory cachexia are not sufficiently explained.

The introduction lacks a clear hypothesis or an open-ended research question that encompasses both the initial characterization and the eventual treatments. As presented and without a hypothesis, the treatments appear more therapeutic in the description, limiting the mechanistic insight. However, this may be more related to the presentation.

Please explain the 30% weight loss euthanasia more clearly, as it may stand out to many readers. 20% weight loss is often the humane endpoint.

For figure 4 and figure 5 experiments, please present the effects of the treatments on the tumor characteristics.

Reviewer #4 (Remarks to the Author):

In this work, Lime Queiroz and colleagues study the cachexia development in the oncogenic KRas- and Stk11 deficiency-driven KL mouse model of lung carcinoma. They show the effects of cachexia mainly in skeletal muscle (glycolytic or oxidative), adipose tissue (white and brown) and blood parameters, and they also perform indirect calorimetry to assess metabolic parameters in mice developing cachexia. They report that KL mice developing cachexia have abnormal WAT, BAT and glycolytic fiber muscle morphology and molecular parameters, as previously reported. Using indirect calorimetry, they observe that cachexia-developing KL mice spend more total energy while moving and eating less, and this effect is not completely rescued by housing them at 30°C. Also, they show that cachexia is improved by a combined treatment with the monoclonal antibody inhibiting Activin ActR1I-Fc and the ghrelin receptor agonist Anamorelin, treatments already described to improve cachexia in mice and humans. This improvement was not observed with combined ActR1I-Fc and the monoclonal antibody inhibiting GDF15. In this work, many relevant experiments are performed, but the presentation and analysis of the data, as well as the conclusions drawn from the observed results, are very confusing and poorly explained, and a major effort must be done before it can be accepted for publication.

These are my comments:

General remarks:

- In general, there is a lack of information about the sex and number of mice used for each experiment/panel. In some panels, individual dots are represented, although the number of these dots are not consistent between different panels of the same figure, and are often hard to count; but very

often no information is given. Please, indicate how many mice are used in each group for each panel.

- The rationale and conclusion of some experiments are not clearly explained. For example, subjecting mice to 30°C (a temperature deemed too high in some reports) did not compensate for weight, TEE, survival and lean mass loss in +CACS mice, but increased activity. The conclusions from this experiment are not explained in detail, and it is unclear what we can infer from these findings: why is increased temperature increasing activity in +CACS mice? Are muscles from +CACS mice housed at 30°C doing better than +CACS mice at 22°C? This effect of temperature on activity is specially interesting, because activity is the most striking difference between +CACS (a deadly condition) and CR (a benign, reversible condition).
- Authors demonstrate a specific sex-dependent effect, however, in the different figures, no mention is made of the sex of the animals in which the experiments have been carried out. Please indicate.
- Authors show that treatment with ActRII-Fc and Anamorelin improve cachexia, which has already been described before. These findings, therefore, lack novelty, and no effort is done to underline any conceptual or practical advance.

There are also an unusually large number of formal or precise comments on inaccuracies, mistakes or unclear panels throughout the manuscript. A strong effort must be done to clarify the manuscript, as it is extremely hard to follow and draw any clear conclusion.

- Figure 1 legend: the phrasing is not clear, revise and clarify please. Also, indicate the sex of the mice in the legend and in Methods section.
 - Figure 1A: explain better what is the difference between +CACS and - CACS mice; it becomes clear as the manuscript advances, but these groups are not properly explained from the beginning. Were both models inoculated with AdCre? What is the comparison marked with the *?
 - Figure S1: describe in the legend the mouse genotypes and treatments shown, as well as the number of mice analyzed for each condition. Statistical methods described in the legend do not correspond with the panels. Also, specify the type of post-tests performed for correction of the statistical analysis, when appropriate.
 - Figure S1C: it seems that -CACS mice have larger adipocytes in WAT and BAT than WT mice. Please, measure adipocyte size and quantify in each group, using a reasonable number of mice and fields to perform proper statistics.
 - Figure S1D: define more clearly what is exactly being shown in the bar graph and the dot graph: number of genes? Batlas Score? % of BAT genes?
 - Figure S1F: quantify IHQ staining. The images are not clear, they look like +CACS mice have larger adipocytes in their BAT, in contrast with FS1C. Also, a higher magnification image of the gWATs would help identifying the UCP1-positive cells.
 - Figure S1G: it is referred to after figure S1H, which is strange. UCP1 protein levels in gWAT are missing in this figure, which is an important measurement since they seem to be highly induced in mRNA, but IHQ is not so clear. Also, from this WB, neither UCP1 nor PGC1 α protein levels look decreased in BAT, at least not consistently. Please, quantify these bands relative to GAPDH (which, by the way, in a metabolically imbalanced situation as cachexia, might not be the most appropriate reference protein; please comment on this point).
 - Figure S1H: what are these mRNA levels relative to? It seems that they are relative to Leptin expression in -CACS mice (value of 1), but this is never indicated. Instead of Student t test, one-way ANOVA should be used for this analysis, since there are several groups being compared. The same applies to Figure S1I and others: when more than two groups are being compared for one single parameter, one-way ANOVA and not Student t test should be used, with the appropriate correction for multiple comparisons.
- Also, Pgc1 α levels are not significantly increased in +CACS gWATs, indicate this in the main text. In this regard, in the text authors specify that "UCP1 expression as increased over 50-fold". Since no protein quantification is shown, the mouse gene expression should be indicated by using italics and lower case ("Ucp1"). As suggested above, protein quantification from IHQ and WB would also be very welcome.
- Figure 1D: this panel is very confusing: the different groups are scattered throughout the graph and the findings indicated in the text are not clearly reflected in it: I do not find +CACS mice being the lowest; instead, I find the 30°C mice being the lowest (both +/-CACS). +CACS at 22°C mice are scattered around, but not necessarily lower than -CACS or Pre-CACS. If anything, +CACS mice have

lower total EE only when lean mass is very low, but not when lean mass is high. This reflects that really cachectic mice (with very low lean mass) have lower EE. I do not think that this graph is the clearest way to say this.

Also, what does the red line represent? All +CACS mice, or only 22°C +CACS mice? This is not clear from the legend. Are the slopes different between +CACS, -CACS and WT (indicating a stronger influence of lean mass in one of the groups)? And does temperature influence this effect/slope?

- Figure S2B: decreased RER is not equivalent to decreased VO₂ consumption; instead, it indicates that metabolism is more reliant on fatty acid consumption than on glucose or aminoacids. A deeper explanation about this decreased RER finding and its implications should be included in the main text when mentioning this finding.

- Figure 1E and 1F: how are food intake and activity in WT mice? What time points are significantly different between groups? 2-way ANOVA tests also give differences for individual points, not only for the whole curve as indicated in the figure.

Also for Figure 1F, it is surprising that mice at 30°C move more than mice at 22°C, I would have expected the opposite, at least for the -CACS mice. Please comment on this finding.

- Figure 1: it is not clear what statistical test is performed for each panel. In the figure legend authors indicate that a 2-way ANOVA test was performed for panels B/D/E, but these graphs represent different types of data and 2-way ANOVA is not appropriate for panel B. Please revise and indicate accordingly.

- Figure S4A and S4B: I understand that the slope of the linear regressions shown in panels A and B are significantly different to 0 in the glycolytic, fast-twitch gastrocnemius, quadriceps and EDL muscles, but no in the slow-twitch, less glycolytic soleus muscle, where $p > 0.05$. However, this is not explained in the text: authors do not mention which are supposed to be the glycolytic/non-glycolytic muscle types or what the slopes are representing. Without this explanation, figures are very hard to follow. Please explain these results in more detail in the text and legends.

- Figure 2B: both LDHA and PDH are glycolytic enzymes that convert pyruvate into lactate (LDH) or Acetyl-CoA (PDH). I understand that PDH drives oxidative phosphorylation for pyruvate, while LDHA drives non-respiratory glycolysis, but no explanation is given in the text, and why LDHA activity is increased in +CACS mice (more pLDHA), while PDHE1a activity is decreased (more pPDHE1a). Please explain this crucial point properly in the text. Also, please quantify these Western blots, normalizing for non-phosphorylated proteins and Tubulin, and provide a statistical analysis of these results.

What happens with these glycolysis/phosphorylation markers in other glycolytic muscles like quadriceps and gastrocnemius (larger than EDL)?

- Figure S5C: by eye, it seems that EDL from +CACS mice have more mitochondria than -CACS mice; while the opposite occurs in the soleus. However, authors report no alterations in the ultrastructural level. A quantification of the surface and evaluation of the morphology of mitochondria would be welcome to support this claim, because the shown EM photographs are not consistent with the text.

- Figure S5D and E: please, explain more clearly what do the codes of the different conditions mean, as they are now explained in the legend I cannot understand them. For example: "NADH-linked (N) LEAK (L) respiration was measured in the presence of pyruvate and malate (PML or PMP)", but I cannot see N or L codes in the figure. Does PM mean "NADH-linked leak respiration"? Does PML mean "NADH-linked leak respiration in the presence of pyruvate (and, if so, why the "L" subindex)?

- Figure 2 and S5: there is a shift between different glycolytic muscle types throughout the figure: soleus and EDL for Figure 2 panels A, B, C, D; gastrocnemius in G, H and I; gastrocnemius in Figure S5 panels A and G; soleus and EDL in panels B-F. Why this inconsistency in the glycolytic muscles used? Justify in the text, or include the analysis for the same type of muscle for all panels.

- Figure 3 and S6: it is not clear how the energy intake in +CACS mice (~8kcal/day) was determined. Also, this energy intake should be normalized to total body mass, because +CACS mice weight less and, therefore, subjecting a heavy, -CACS mouse to this drastic CR can be much more stressful than doing it to a thin +CACS mouse.

- Figure 3D: the units are not represented in the figure.

- Figure S6C: as in Figure S2B, an explanation of the metabolic implications of the reduced RER in CR mice should be included in the text. Also, there is no statistical difference, however, it is indicated in the manuscript.

- Figure S8C and D: it is very hard to assess most findings described in the text by the images shown in these figures, (reduced fiber size or UCP1 staining), since the magnification is too low and there is not enough detail. Also, to robustly support these descriptions, a quantification of several fields from several mice of each group, with appropriate statistics, should be performed.
- Figure 4: authors affirm that the combined therapy protects against the loss of glycolytic skeletal muscle, however, the increase in the amount of these tissues is very low. Please, include data of -CACS mice to verify if this protection is comparable to -CACS mice.
- “Anamorelin hydrochloride” has been approved as a treatment for CACS. Why is “anamorelin fumarate” (Ana) used in the present manuscript? If authors are trying to compare “Anamorelin fumarate” against an already approved treatment, please indicate, or discuss this difference.
- Figure 4A and B: statistics should be performed using the one-way ANOVA (4 groups, one parameter), not the indicated two-way ANOVA.
Are +CACS mice significantly different from -CACS mice in these serum parameters?
Also, when describing the antibody treatments, please explain what “QW, SQ” means.
- Figure 4D: it is extremely striking that the difference in weight loss between start and end of treatments is significant in the combination group (where some mice increase and others decrease their weight), while it is not significant in the other groups, with more consistent and stronger changes. Please revise this statistical assessment.
- In the S11A and S11B panels, authors indicate that the Ana + ActRIIB-Fc treatment significantly increases food intake and weight loss after two weeks. However, the animals treated only with Ana (a drug already used against CACS) did not show any increase compared to the IgG group. Is the Ana dose and regimen used appropriate in this model? Please comment.
Figure S11H: please, specify the meaning of the different signs for statistical significance: * and #.
The figure legend is not clear.
- Figure S12: it is not clear how many male mice were analyzed. In many figures, only one dot is represented (C, D, E). Please, specify how many male mice are included in this assay. If they were too few, as it seems from some of the graphs, the negative conclusion reached by authors for male mice are not robust enough and should be commented in the text and Discussion section.
- Figure 5G: authors claim that combination therapy in females increased overall survival; but there is no sign indicating significance in this graph. If combination therapy-treated mice live longer, indicate significance; otherwise, restate these finding in the text (both in Results and in Discussion), indicating a “clear tendency” towards longer lifespan.

Methods:

- Please, indicate the sequences of the primers used for qPCRs.
- Describe how the CACS- and CACS + animals have been identified.
- The Caloric Restriction experiments carried out in this manuscript are not described in the methods section.

In general, the work by Lima-Queiroz and colleagues is extense and deals with a clinically relevant condition. However, the presentation of results is very confusing, and the actual relevance and novelty of the findings to the field of research is not very convincing, and should be strongly improved to be considered for publication in a journal of the relevance of Nature Communication.

Focused responses to the reviewers

Reviewer #1:

Cancer-induced cachexia clearly represents a major unmet clinical need with currently no pharmacological therapy approved. In this regard, the study by Lima Queiroz et al. addresses an important and timely topic. Overall, the study employs state-of-the-art technology, appropriate models and is clearly structured and well written.

Thank you very much.

The reported reduction in systemic energy expenditure is interesting as it contradicts other studies in the field. However, these findings have been reported previously (Rohm et al. Nat Med. 2016), and thus lack a certain degree of novelty.

We thank the reviewer for highlighting the previously published data and we have now cited it appropriately in the text. In Supplementary Figure 4 of Rohm *et al.* 2016, the authors nicely show that mice bearing C26 allografts display reduced oxygen consumption and resting metabolic rate (RMR) at a late stage, as compared to pair-fed mice. Interestingly, these reductions occur without changes in food intake – unlike the KL model used in our study. In fact, the pair-fed, non-tumor-bearing mice from Rohm *et al.* do not lose weight over time, suggesting that the provided food is enough to match the total energy expenditure of WT mice. Based on this data, we speculate that the reduction in RMR in the C26 model is due solely to changes in peripheral metabolism and combination therapy (targeting appetite and peripheral metabolism) is not needed. Our manuscript nicely complements their study, highlights the heterogeneity of phenotypes that are found in cachexia, and addresses the need for multimodal therapy in this complex condition.

The manuscript in general remains at a rather descriptive level. In particular, the mechanisms of gender-specific responses to pharmacological interventions are interesting and should be explored in more mechanistic details.

We agree that the mechanism of the sex-specific response is interesting and have addressed several hypotheses in the revised manuscript.

- First, we tested whether Activin A was only upregulated in female mice, and this was not the case. Both male and female mice have equally high levels of Activin A.
- Next, we tested whether the beneficial effects of the Ana-ActRIIB-Fc combination intervention in female mice were dependent on estrogen receptor (ER) activity. We performed a 3-arm prospective trial where cachectic female mice were randomized to receive vehicle, fulvestrant (an ER degrader), or fulvestrant plus the combination therapy. Fulvestrant did not limit the

effects of the combination therapy on weight, fat mass, or lean mass suggesting these effects are independent of estrogen signaling.

- We considered the endocrine role of Activin A, which is known to stimulate FSH production by the pituitary gland and enhance granulosa cell proliferation in the ovaries. We measured FSH levels in female mice with or without cachexia and found no change. Furthermore, we reviewed the histology of the ovaries from mice with or without cachexia and those with cachexia treated with the combination therapy. There was a trend for the cachectic ovaries to be smaller and the treatment tended to correct this change; however, this analysis was underpowered.
- Lastly, we performed a study to test the role of ovarian hormones in the response to the combination therapy. KL mice underwent ovariectomy (n= 21) or a sham surgical procedure (n= 20) prior to induction of lung cancer. The mice were then randomized to receive vehicle, Ana, or Ana+ActRIIB-Fc once they developed cachexia. While mice that underwent sham surgery regained weight with the combination treatment as expected, this was not the case for the ovariectomized mice. These data suggest that an ovarian factor that does not signal through the estrogen receptor is a critical mediator of the combination therapy. We speculate that an ovarian androgen is important for this response, and further studies will delineate this factor.

Figure 1 The beneficial effects of the combination therapy are abolished by ovariectomy but not by estrogen receptor inhibition. (A) Serum levels of Activin A stratified by sex in mice without (-CACS, n=12) and with CACS (+CACS, n=15) mice. (B) Food intake of female control mice (n=6), fulvestrant treated mice (n=4), and mice treated with fulvestrant in combination with Ana and ActRIIB-Fc decoy mAb (n=4). (C) Weight loss of female control mice (n=6), fulvestrant treated mice (n=4), and mice treated with fulvestrant in combination with Ana and ActRIIB-Fc decoy mAb (n=4). Change in (D) lean mass and (E) fat mass of control mice (n=6), fulvestrant treated mice (n=4), and mice treated with fulvestrant in combination with Ana and ActRIIB-Fc decoy mAb (n=4). (F) FSH levels in the serum of -CACS and +CACS female mice. (G) Cross sectional area of the ovaries of NCACS (n=2), CACS (n=6) and CACS mice treated with Ana+ and ActRIIB-Fc decoy mAb (n=4). (H) Total body weight loss at Start and End in mice that underwent sham surgery treated with either Control (n=4), Ana (n=4) or Ana together with an ActRIIB-Fc decoy mAb (n=5). (I) Total body weight loss at Start and End in mice that underwent ovariectomy treated with either Control (n=5), Ana (n=5) or Ana together with an ActRIIB-Fc decoy mAb (n=4).

The relevance of the reported findings for the human pathology remain unclear

To address this concern, we examined publicly available clinical databases and included our main findings in Figure 6 of the new manuscript. Briefly, we find that:

1. **Activin A protein abundance is increased in 2 distinct proteogenomic subtypes of human lung cancer that have poor survival (Fig. 6C) (Lehtio et al., 2021).**
2. **The lung cancer dataset of the Clinical Proteomic Tumor Analysis Consortium (CPTAC) includes body mass index (BMI). Since a BMI <20 kg/m² has high specificity for cachexia (Fearon et al., 2011), we compared the relative protein levels of Activin A in the tumors of patients with BMI >20 compared to BMI<20 and found it significantly increased in the latter (Fig 6 A).**
3. **High levels of *INHBA* expression correlate with poor survival only in female patients with lung adenocarcinoma within the TCGA (Fig. 6E-G).**
4. **Activin A protein levels are elevated in the serum of lung cancer patients from Weill Cornell.**
5. **Reductions in skeletal muscle oxidative capacity have been observed across a range of mouse models and clinical biopsy samples. Nicely summarized by Rogier *et al.* (Rogier et al., 2021).**

Specific Comments

1. Please explain the rationale for using exactly these pharmacological combination therapies (amongst all other possible combinations, and beyond their simple availability). The use of the combination therapies is not deducted from any specific data obtained by the authors, and thus it remains unclear why this road was taken.

In Figure 4, we now include additional data that led us to use the ActRIIB-Fc intervention to target skeletal muscle atrophy and metabolism. Specifically, *Inhba* was identified as differentially upregulated in tumors from cachectic mice as compared to those from non-cachectic in an unbiased RNA-Seq analysis. In the skeletal muscle, the TGF- β signaling pathway was identified as an enriched pathway by GSEA analysis of the RNA-Seq. We then validated that Activin A was specifically increased in the serum of mice with cachexia.

The regulation of appetite during cachexia is an emerging field with a limited number of well-characterized anorexic factors (Olson et al., 2021). Of these factors, GDF-15 was found to be increased in the blood of mice with cachexia and this led us to specifically block GDF-15 activity with the anti-GDF-15 antibody. When that was unsuccessful, we turned to anamorelin, a drug that is known to stimulate appetite in humans and mice with cancer. We were pleased to see that it worked well in our model.

Reviewer #2:

Overall, the study by Queiroz et al is comprehensive, addresses an important biological problem, and relevant therapeutics, and provides some novel insights into cachexia pathophysiology.

Thank you very much.

According to the AAALAC guidelines for welfare and use of animals in cancer research, animals should not be allowed to exceed 20% weight loss without euthanasia. In this study, it appears that many animals went beyond this, with some losing nearly 40% of body weight (figure S1a, figure 5b). Please include a statement about the exception to the AAALAC guidelines, and how humane endpoints were monitored in the study. It is worth discussing the implications of such severe weight loss in the experimental animals- this is well beyond any cachexia definition in humans and is an extreme physiological state. Indeed, in the drug trials, the animals had already passed 15% weight loss before initiation of therapy. Is it possible this represents “refractory cachexia” given the severe weight loss in this cohort? Finally, were the tumor masses in these animals recorded?

We acknowledge the concern for the degree of weight loss observed in our study and appreciate the opportunity to address this important topic. Our study was performed with the approval of the Institutional Animal Care and Use Committee (protocol 2013-0116), which agreed to 30% weight loss as a humane endpoint. In a prior study (Goncalves et al., 2018), we characterized the metabolic changes that occur in KL mice with and without cachexia. We found that systemic metabolic dysfunction, anorexia, and muscle atrophy only occurs after 15% weight loss. Therefore, this threshold is used as the defining criteria for CACS in our model. As shown in Figure 1A (reproduced on the right), the mice reach 30% weight loss 7-14 days after the onset of CACS. This period was determined to be an adequate duration for therapeutic intervention and for the full development of the cachectic phenotype.

The tumor bearing mice in our trials are monitored weekly for weight, food intake, and body composition score (a score <2 is another humane endpoint). When mice reach 20%, they are monitored and weighed daily. In some instances, mice rapidly lose weight and exceed the 30% weight loss criteria at the next measurement. Those mice are euthanized immediately. This information has been added to the methods in a new **Humane Endpoints** section.

We refrain from using the term “refractory cachexia” since it is unclear when this phenomenon occurs in our model. It is clear that 15% weight loss is not refractory since the combination of Ana+ActRIIB-Fc reverses weight loss (Figure 5G, reproduced on the right). Nevertheless, we have included this point as a limitation of our study in the discussion on page 22.

Total lung mass is used as a surrogate for tumor burden in this model. This data is now presented in Figure 1D (reproduced on the right) and Supplemental Figures 5A and 5E.

Although a great deal of the data in this paper is dependent on accurate measurement of food intake, no description of the methods for determining food intake are provided. Were animals individually housed? Was food intake automated? Solid food or powdered food? Was bedding screened for food waste? The latter is critical as it is evident from multiple prior studies that food wasting and shredding is a common behavior in sick mice.

We apologize for this oversight. In Figure 1, food intake was measured in real time from single housed mice using high precision sensors as part of the Promethion food intake monitoring system. The food pellets are crushed and placed into a stainless-steel container, which sits atop the sensor. There is a built-in 'crumb catcher' tray that eliminates spilling and reduces caching behavior, providing more accurate and repeatable results.

In Figures 5A and 5F, food intake was measured manually from single housed using a digital scale. The food pellets are left intact in the overlying hopper and the mass is recorded 3 times per week approximately between 10-12 am. The residual food powder at the bottom of the cages is collected by sifting out the bedding using a baking sifter; however, this amount is never significant unlike other models. This information has been added to the methods.

In figure 1c,f the -CACS animals have much higher activity at 30C than at 20C, yet the energy expenditure at 30C is much lower than at 20C. Presumably this is because of a difference in BMR. It would be helpful to estimate the difference in BMR under these two conditions, and put that into the context of what is already known about the impact of ambient temperature on metabolic rate in mice.

Indeed, the increase in spontaneous activity of non-cachectic mice at 30°C is associated with a reduced resting energy expenditure (below). This phenomenon has been previously observed (Škop et al., 2020).

Figure 2 Resting Energy Expenditure (REE) in non-cachectic (-CACS) KL mice at 22 and 30°C. REE was estimated using a previously described algorithm (Van Klinken et al., 2012) that utilizes spontaneous activity and total energy expenditure measures obtained from metabolic chambers. Animals were single-housed mice at 22°C (n=15) and 30°C (n=12). Comparison made via t-test. ****P<0.0001

Mice at typical housing temperatures (20°C–22°C) live below thermoneutrality, and about half of their total energy expenditure is devoted to maintaining core body temperature. Increasing the housing temperature reduces the need for thermogenesis and, consequently, the resting energy expenditure falls. We speculate that the reduced requirement for thermogenesis makes more energy available for spontaneous activity but the mechanisms controlling spontaneous activity in mice are unknown.

The RER data are also a bit puzzling- the food intake of the +CACS 22C and -CACS 30C are the same, yet their RER is quite different. How might this be explained?

RER is an indirect readout of whole-body substrate oxidation that is influenced by numerous exogenous and endogenous factors such as: the level of feeding (positive vs. negative energy balance), the size of the glycogen stores, insulin sensitivity, and the amount of lipid in the adipose tissue (Schutz, 1995). The cachectic mice at 22°C are in a state of negative energy balance with depleted white adipose tissue depots, defective hepatic fatty acid oxidation, and active glucose oxidation in the brown adipose tissue for thermogenesis. The non-cachectic mice at 30°C are in neutral energy balance with normal adipose tissue reserve, normal hepatic fatty acid oxidation, and no excessive glucose oxidation needed for thermogenesis. So, while the food intake is the same in these two conditions, the metabolic state of the animals is very different.

The method of dividing mice into +CACS and -CACS is confusing. Presumably, this is done retrospectively, with animals reaching a 15% weight loss being the +CACS group. Review of figure 1A suggest that in fact all mice are losing weight, with the -CACS “late” group being at or below the +CACS “mid” group. Isn’t this just a reflection of the stage of CACS development? That is, the rate at which the +CACS group is losing weight is just a bit faster than the -CACS group, perhaps due to differences in the timing, degree, and anatomical location of the tumors?

Yes, the presumption is correct. A cohort of KL mice is induced with adenovirus and then the animals are monitored over time. Those that reach 15% weight loss are defined as +CACS. 15% weight loss is the point where systemic metabolic dysfunction develops in this model (Goncalves et al., 2018). We have clarified this distinction in the methods.

To date, we have completed 29 cohorts of KL mice with up to 60 mice per cohort and, in our experience, the -CACS will never reach 15% weight loss in their lifetime. We have followed the mice for up to 20 weeks after induction. Those that do not develop CACS are typically found dead in their cages without significant alteration in body weight. We presume this death is related to infection, hemorrhagic and thromboembolic phenomena, and respiratory failure; however, it has not been thoroughly investigated. We have also performed serial MRI imaging on the KL mice and found that all mice develop tumors by 4 weeks after induction regardless of their impending CACS status. We now provide a new version of Figure 1A showing the fluctuations in body weight over time to highlight that only the +CACS mice lose weight.

The reviewer brings up an interesting point about the anatomic location of the tumors. We have preliminary data to suggest that tumor location and tumor histologic subtype influence the CACS phenotype. In ongoing experiments in KL mice with conditional Cas9 expression, we are using lentivirus to deliver Cre recombinase and specific guide RNAs to the lung in an attempt to manipulate tumor subtype and determine the effects on CACS readouts.

What is the final tumor mass of these various groups? For RNA seq studies, what stage of weight loss were the +CACS animals relative to the -CACS? If end-stage, with a minimum of 15% weight loss, then the limitations of interpretation of these data should be discussed. Specifically, this would represent severe, refractory cachexia in a human, and observed changes in gene expression are more likely to represent that stage than to represent causal, driving pathways that would have been active much earlier in the cachexia process.

The final tumor mass is presented as total lung mass, a surrogate for tumor burden in the KL model.

Lung mass is similar between mice with and without CACS. These data have been added as Figure 1B (reproduced above).

For RNA Seq, the tissues were harvested at study endpoint. We agree that this approach cannot distinguish primary, causal pathways from the secondary response. This limitation is now described in the Discussion on page 22.

Lung cancers in humans, and previous studies with lung cancer models in mice show increased levels of PTHrP in circulation, and at least one study implicates this protein in the browning of WAT. If samples remain available, it is therefore reasonable to assay the levels of this endocrine molecule in these studies.

We agree that PTHrP is a viable candidate for the browning effect observed in our model and measured its abundance in serum and in tumor lysates from -CACS and +CACS mice. Although we observed high levels in the tumor lysates (below), PTHrP was undetectable in the serum.

Figure 3. Concentration of PTHrP in tumor lysates of KL mice.

There were two females in the GDF15 study that apparently responded to treatment. Is anything known about their tumor burden? Did they have improvements in activity and food intake?

These mice had normal amounts of tumor burden and food intake. One of the mice displayed extreme levels of cumulative activity (>1000 meters per day), which is likely a statistical outlier. The other mouse had normal physical activity levels.

Figure 4. Additional Results from GDF-15 intervention trial. (A) Lung mass, (B) cumulative daily activity, and (C) food intake over time from mice in the GDF-15 intervention trial. Arrows depict the two female mice who responded to treatment.

In S11 and S12, the numbers of animals in the treatment arms does not seem to match the number analyzed in the fat and lean mass (c, d in each figure). For example, there is only a single animal point in figure S12 c,d for the combined therapy. Why are the data for the other animals not shown? Similarly, in figure 5, the numbers in the data for ending weight do not match the numbers for the fat mass and lean mass and one of the dots for weight loss percent in the combined treatment group does not have a connecting line to a starting weight. The numbers in the lung masses (Fig 5e) also do not line up with the numbers in the lean and fat mass.

Thank you for the opportunity to address this issue. The KL cohorts are not easy to perform and require constant surveillance of the mice and scheduling with our Mouse Metabolic Phenotyping Core that performs the body composition and indirect calorimetry. Following induction of tumorigenesis in the KL cohorts, the data points for each mouse are acquired at different times depending on the degree of weight loss. Sometimes, mice are found dead in their cages or reach humane endpoints before body composition or indirect calorimetry can be scheduled/performed. When mice are found dead, we do not harvest the tissues and cannot obtain lung mass. For the experiment presented in the former Figure S12, we captured food intake and changes in body weight for all mice but were only able to obtain body composition and indirect calorimetry data from one male mouse. The others died too quickly. Similarly, in Figure 5, not all mice had body composition performed. We included a description of our study approach in the “**Therapeutic trials**” section of the methods.

For clarity, all the figure legends have now been revised to include the total sample size for each group of each panel.

Thank you for pointing out the error in the mouse that did not have a matching end weight. All panels have now been reviewed, confirmed, and corrected.

The study of combined Ana/ActRIIb was divided into sex post hoc. Was this study adequately powered to address this variable? In general, the dangers of post hoc analysis based on a single experiment should be addressed in the discussion.

In the initial design of the GDF-15 and Ana studies, we used prior data to determine that 7 mice per group were required to detect a 20% change in mean weight loss ($\alpha=0.05$, $\beta=0.9$). Additional mice were added to account for early mortality and the proportion of -CACS mice. Following induction, if mice lost 10-15% of their peak weight, they were randomized to a treatment arm. The randomization was performed in blocks of 6 and stratified by sex (Pocock, 2013). The pre-specified primary outcome was the percent weight loss at 2 weeks following the start of treatment (End). If a mouse met endpoint criteria (loss of 30% body weight or body composition score <2), before the End date, then -30% would be used for their End value. Secondary outcomes included food intake, overall survival, body composition, food intake, spontaneous activity, skeletal muscle mass, and white adipose tissue mass. Overall survival was calculated from the date of induction. For the GDF15 trial, 39 mice (22 males, 17 females) were induced, and 29 mice (17 males, 12 females) reached 15% weight loss and underwent randomization. After we observed the increase in body weight from GDF15+ActRIIB-Fc in the two female mice in this cohort, we planned to increase the number of female mice in the Ana cohort to ensure adequate power. Specifically, a total of 50 mice (26 males, 24 females) were induced in the Ana cohort. In this particular cohort, 46 mice (23 males, 23 females) reached 15% weight loss and underwent randomization. These details have been added to the “**Therapeutic trials**” section of the methods.

Moreover, we also prospectively reproduced the improvement in female body weight in 2 new cohorts where the effects of fulvestrant and ovariectomy were investigated. Less animals were used in these follow up studies since the effect size was so large in female mice.

Figure 5. Ovariectomy but not estrogen receptor inhibition prevents the beneficial effects of the combination therapy. (A) Serum levels of Activin A stratified by sex in mice without (-CACS, n=12) and with CACS (+CACS, n=15) mice. (B) Food intake of female control mice (n=6), fulvestrant treated mice (n=4), and mice treated with fulvestrant in combination with Ana and ActRIIB-Fc decoy mAb (n=4). (C) Weight loss of female control mice (n=6), fulvestrant treated mice (n=4), and mice treated with fulvestrant in combination with Ana and ActRIIB-Fc decoy mAb (n=4). Change in (D) lean mass and (E) fat mass of control mice (n=6), fulvestrant treated mice (n=4), and mice treated with fulvestrant in combination with Ana and ActRIIB-Fc decoy mAb (n=4). (F) Food intake of female mice that underwent sham surgery or ovariectomy and were then randomized to receive vehicle (Sham-Control, n=4; Ova-Control, n=5), or combination therapy with Ana and ActRIIB-Fc decoy mAb (Sham-Ana+ActRIIB-Fc, n=6; Ova- Ana + ActRIIB-Fc, n=5). (G) Weight loss of female mice that underwent sham surgery or ovariectomy and were then randomized to receive vehicle (Sham-Control, n=4; Ova-Control, n=6) or combination therapy with Ana and ActRIIB-Fc decoy mAb (Sham-Ana+ActRIIB-Fc, n=6; Ova- Ana + ActRIIB-Fc, n=5). Change in (H) lean mass and (I) fat mass of mice that underwent sham surgery or ovariectomy and were then randomized to receive vehicle (Sham-Control, n=4; Ova-Control, n=5) or combination therapy with Ana and ActRIIB-Fc decoy mAb (Sham-Ana+ActRIIB-Fc, n=6; Ova-Ana + ActRIIB-Fc, n=5). Only female mice were used in panels B-I. Graphs show mean \pm SEM. Comparison in A was done using unpaired Student's t-test while comparisons in B/C/F/G were done using paired t-test. D/E/H/I were analyzed using one-way ANOVA. * $p < 0.05$.

The “loss of insulation” hypothesis raised in the third paragraph of the discussion is interesting-how is this consistent with the observations in pure calorie restriction? Does this imply that no browning should be observed in animals in a thermoneutral environment?

The reviewer brings up a good point that challenges our hypothesis. Following calorie restriction (CR), mice lose similar amounts of weight (or insulation) and should have equal amounts of WAT browning. To test this hypothesis, we quantified the UCP1 positive area in the gonadal WAT depot following CR. The CR cohort had significantly increased UCP1 staining, however this difference was relatively small in magnitude compared to mice with CACS. We have removed the “loss of insulation” hypothesis from the discussion.

Figure 6. UCP1 staining in WT mice in the fed state and following calorie restriction (CR). Reproduced from Figure S1. (J) Immunohistochemistry for UCP1 was performed on gonadal white adipose (gWAT) or brown adipose (BAT) tissues using a validated antibody and representative images are shown. (K/L) Quantification of UCP1 positive area from the gWAT and BAT in J. Comparisons by t-test. *P<0.05

The reviewer's intuition is correct regarding the browning at thermoneutrality. We did not find any evidence of UCP1 staining in the adipose of tumor-bearing mice housed at 30°C; however, the sample size is too small (n=5 per group) to make any firm conclusions. We include the following statement on page 7: "We were unable to detect any histologic evidence of browning in the cachectic gWAT when mice were housed at 30°C; however, the number of mice with available tissue was small (n=5 per group) and this limitation affected our ability to make a definitive conclusion (data not shown)."

For the forced treadmill studies, it is important to point out that mice are far more limited by cardiac output than humans are, and cardiac reserve and heart function may be more limiting in this study than skeletal muscle function. Any information on heart muscle would be useful to include, if available. Furthermore, cardiac function as a potential driver on mobility should be added to the discussion.

We agree that cardiac output may limit the aerobic capacity of +CACS mice. We did not perform assessments of cardiac function; however, we did not see any significant changes in heart mass across the cohorts. We added a statement acknowledging the lack of proper cardiac output measures as a limitation in the Discussion. This data has not been included in the manuscript.

The statement is made that the BAT was smaller in the CACS mice, but that the size of the adipocytes did not change. How is this reconciled?

We show that the total BAT mass is reduced in mice with CACS as compared to those without CACS (Figure S1A). To address the source of the lost weight, we quantified the amount of fat in the BAT using unbiased image quantitation performed by blinded investigators. We found that the mean fat content in the BAT is reduced in the cachectic mice, however this measurement was highly variable, and the difference did not reach statistical significance (Figure S1L/M reproduced below). We also quantified the UCP1 positive staining in the BAT since our initial impression was that the mice with CACS had lower staining intensity and the UCP1 gene expression in the BAT was lower than in -CACS. Again, the measure was highly variable with clusters of mice showing high and low UCP1 positivity (Figure S1N below). The UCP1 low clusters did not correspond to any specific cohort, sex, weight loss, or tumor burden.

Figure 7. Heart mass from KL mice. Groups were compared by one-way ANOVA: nonsignificant.

Figure 8. histologic effects of CACS on brown adipose tissue (BAT). Reproduced from Figure S1. (L) Representative H&E (top) and UCP1 immunohistochemistry (bottom) staining of the BAT from WT, -CACS, and +CACS mice. (M/N) Quantification of fat area and UCP+ staining from L. Five high powered fields (20x) were selected randomly from WT (n=4), -CACS (n=5) and CACS (n=5) mice. Comparisons in M and N made by one-way ANOVA followed by Tukey's multiple comparisons test *P<0.05

In contrast to the results from mice with CACS, the amount of fat in the BAT is significantly reduced in mice that have undergone CR (Figure S4F-H, reproduced below). These data highlight the distinct metabolic changes that occur in CACS independently of reduced food intake.

Figure 7 histologic effects of CACS on brown adipose tissue (BAT). Reproduced from Figure S1. (L) Representative H&E (top) and UCP1 immunohistochemistry (bottom) staining of the BAT from WT, -CACS, and +CACS mice. (M/N) Quantification of fat area and UCP+ staining from L. Five high powered fields (20x) were selected randomly from WT (n=4), -CACS (n=5) and CACS (n=5) mice. Comparisons in M and N made by one-way ANOVA followed by Tukey's multiple comparisons test *P<0.05

Gender is not applicable to mice- this should be changed to “sex” throughout. Although the issue of gender specificity of pathology and therapeutic response is interesting, what is being studied here is sexual dimorphism.

We agree with the reviewer and have rephrased the manuscript text changing “gender” to “sex”.

Under “metabolic cage analysis” the sentence implies that both O₂ and CO₂ are “consumed”. Presumably you mean “produced” for the CO₂.

We agree with the reviewer and have rephrased the manuscript text changing “consumed” to “produced” after mentioning CO₂.

Need to confirm that only a single transcript was amplified in the qPCR studies, particularly because this is done with SYBR rather than a primer/probe pair.

Only Ppargc1a, RER1 and Rpl41 were amplified with SYBR and we have added the primer sequence to Table 3. These sequences were blasted to guarantee the specificity of the amplification product and had been previously validated by others (Perez et al., 2017). The rest of the transcripts were measured using Taqman Probes.

It is currently unclear throughout the manuscript under what thermal conditions the various studies were performed. The materials and methods just states “at 22 or 30 C” but does not state when these two different conditions were used.

We understand the reviewer’s concern and have updated the methods to clarify where these conditions were used. We also updated the legends where these conditions were used.

In figure 3A, the Y axis should read “percent baseline weight”, not “% weight loss”.

Figure S8A Y axis units should be mg not g.

Both corrected.

Reviewer #3

The clearly proposed but undervalued descriptive premise for investigating the regulation of energy expenditure with cancer cachexia is still strong and important.

Overall, the study is comprehensive, contains valuable information, and is very well conducted. Important information is abundant for the cachexia field.

Thank you.

The primary issue with impact is that almost none of this novelty can be grasped from the introduction, which reads as a much more simplistic descriptive study. The nuances of sex, caloric restriction, successes, and failures of the treatments used, the novelty of the model, and the design that incorporates treatment are left for the reader to struggle with and discover.

We appreciate this feedback. The manuscript has been rewritten to highlight the novelty of the model, rationale of the study, and impact on the field.

The fact the overall presentation resonates as model-specific serves to lower the potential impact.

Cachexia is a heterogenous condition and it remains unclear which patient subsets are represented by each pre-clinical model. In the updated manuscript, we present new clinical evidence to highlight the translational value of our findings using TCGA, CPTAC and a new proteogenomic classification of lung cancer published by Lehtio et al. (Lehtio et al., 2021). The KL mice appear to recapitulate subtypes 4 and 6 from Lehtio et al. and we suspect cachexia is enriched in these populations. Despite modeling only specific human lung cancer subtypes, we believe the basic principles around anorexia, energy expenditure, and peripheral metabolic dysfunction are applicable to all pre-clinical models and patients with cachexia. In the rewritten manuscript we have better emphasized the applicability of our findings.

There is far too much supplemental data, which is not sufficiently explained, seemingly being a data dump so that it can be called "published." This should be substantially reduced to the data that can be sufficiently explained in the manuscript and provide direct support.

We acknowledge this criticism. Our intention was to comprehensively phenotype the many tissues impacted by cachexia; however, we agree with the Reviewer and have removed and consolidated the data presented.

The introduction carefully defines the metabolic terms and lung cancer model. Still, it is superficial in detailing the treatments and inhibitors described in the introduction's lengthy results portion. Still, the premise related to what is known and their function is insufficient.

We thank the Reviewer for this specific feedback. We have substantially revised the introduction and results based on the Reviewer's critique.

The equivocal role of anorexia in several preclinical models of cachexia is not clearly defined.

We have included a comment in the introduction to acknowledge this point.

Furthermore, there is a current emphasis on the initiation and reversal of cachexia in the design, but not in the rationale. The mechanisms investigated early drivers cachexia initiation or limited to refractory cachexia, which is not clearly explained.

We now note two important limitations of our work in the discussion. “First, our tissue analyses were limited to samples obtained from the studies’ endpoint, a time where numerous metabolic pathways have been perturbed. Therefore, we are unable to distinguish the primary, causal events from the secondary response. Second, our intervention trials enrolled mice that lost 15% body weight. Our prior work suggests that this threshold marks the appearance of systemic metabolic dysfunction, and our goal was to treat existing CACS as opposed to preventing mice from developing CACS. Nevertheless, this amount of weight loss may be considered “refractory” in other models.”

Clearly, 15% weight loss is not refractory in KL mice as the combination of Ana+ActRIIB-Fc reverses it.

The results in figure 1 are confusing related to early and late. Please clarify the presentation of repeated measures in the same mice, which seems to be a strength of the design.

We modified the presentation of Figure 1 for clarity (reproduced below). The individual growth curves from each mouse are now superimposed over the early, mid, and late timepoints.

Figure 8. Updated Figure 1 panels A through D.

The possibility that cachexia can be initiated and metabolic dysfunction occurs before many classic signs of overt refractory cachexia are not sufficiently explained.

Since the KL mice respond well to the combination therapy, we withhold using the term “refractory cachexia”. Nevertheless, we agree with the Reviewer that cachexia is initiated long before the onset of weight loss. In this study, our goal was to *treat* cachexia as opposed to *prevent* cachexia so our phenotyping and analyses were done after weight loss was initiated. In ongoing work in the lab, we are searching for early, pathophysiologic drivers of cachexia by screening tumor and serum samples collected 4 to 5 weeks after tumor induction when mice already have detectable tumors evaluated by MRI but not significant weight loss. These factors could then be targeted to prevent cachexia.

The introduction lacks a clear hypothesis or an open-ended research question that encompasses both the initial characterization and the eventual treatments. As presented and without a hypothesis, the treatments appear more therapeutic in the description, limiting the mechanistic insight. However, this may be more related to the presentation.

We greatly appreciate this feedback. We now more clearly state the hypothesis in the introduction: “In this study, we hypothesized that anorexia and hypermetabolism would contribute to negative energy balance in CACS and used the KL model to interrogate the changes in food intake, peripheral organ metabolism, and EE that occur following the induction of lung cancer.”

Please explain the 30% weight loss euthanasia more clearly, as it may stand out to many readers. 20% weight loss is often the humane endpoint.

We acknowledge the concern for the degree of weight loss observed in our study and appreciate the opportunity to address this important topic. Our study was performed with the approval of the Institutional Animal Care and Use Committee (protocol 2013-0116), which agreed to 30% weight loss as a humane endpoint. In a prior study (Goncalves et al., 2018), we characterized the metabolic changes that occur in KL mice with and without cachexia. We found that systemic metabolic dysfunction, anorexia, and muscle atrophy only occurs after 15% weight loss. Therefore, this threshold is used as the defining criteria for CACS in our model. As shown in Figure 1A, the mice reach 30% weight loss 7-14 days after the onset of CACS. This period was determined to be an adequate duration for therapeutic intervention.

The tumor bearing mice in our trials are monitored weekly for weight, food intake, and body composition score (a score <2 is another humane endpoint). When mice reach 20%, they are monitored and weighed daily. In some instances, mice rapidly lose weight and exceed the 30% weight loss criteria at the next measurement. Those mice are euthanized immediately. This information has been added to the methods in a new **Humane Endpoints** section.

For figure 4 and figure 5 experiments, please present the effects of the treatments on the tumor characteristics.

These data have now been included in Figure S5 A and E. There was no change in tumor burden.

Reviewer #4:

In this work, many relevant experiments are performed, but the presentation and analysis of the data, as well as the conclusions drawn from the observed results, are very confusing and poorly explained, and a major effort must be done before it can be accepted for publication.

We have greatly updated the manuscript to make the experiments and their rationale clearer. General remarks:

- In general, there is a lack of information about the sex and number of mice used for each experiment/panel. In some panels, individual dots are represented, although the number of these dots are not consistent between different panels of the same figure, and are often hard to count; but very often no information is given. Please, indicate how many mice are used in each group for each panel.

Thank you for the opportunity to address this issue. The KL cohorts are not easy to perform and require constant surveillance of the mice and scheduling with our Mouse Metabolic Phenotyping Core that performs the body composition and indirect calorimetry. Following induction of tumorigenesis in the KL cohorts, the data points for each mouse are acquired at different times depending on the degree of weight loss. Sometimes, mice are found dead in their cages or reach humane endpoints before body composition or indirect calorimetry can be scheduled/performed. When mice are found dead, we do not harvest the tissues and cannot obtain tissue masses. We now include a detailed description of our study approach in the “**Therapeutic trials**” section of the methods.

For clarity, all the figure legends have now been revised to include the total sample size and sex for each group of each panel.

- The rationale and conclusion of some experiments are not clearly explained. For example, subjecting mice to 30°C (a temperature deemed too high in some reports) did not compensate for weight, TEE, survival and lean mass loss in +CACS mice, but increased activity. The conclusions from this experiment are not explained in detail, and it is unclear what we can infer from these findings: why is increased temperature increasing activity in +CACS mice? Are muscles from +CACS mice housed at 30°C doing better than +CACS mice at 22°C? This effect of temperature on activity is especially interesting, because activity is the most striking difference between +CACS (a deadly condition) and CR (a benign, reversible condition).

We apologize for the lack of clarity.

The following rationale has been added to page 7 in the text:

“Mice need to actively generate heat to maintain body temperature under standard housing conditions (22°C). This adaptive thermogenesis increases REE and may disproportionately contribute to weight loss during CACS. To assess the contribution of cold-induced thermogenesis to CACS, we performed a prospective, randomized, controlled trial (RCT) where KL mice were induced with AdCre at 22°C and then randomly assigned to stay at 22°C or move to a thermoneutral temperature (30°C) four weeks after induction, which allows enough time for normal tumor development. When mice are housed at 30°C, there is minimal contribution of cold-induced thermogenesis to total EE (Abreu-Vieira et al., 2015; Tschop et al., 2011).”

The main conclusion of the study is that higher ambient temperature, and subsequent relief of thermoregulatory EE, did not alter the progression or development of CACS in KL mice. There was no difference in body weight loss, muscle mass, WAT/BAT mass, TEE, or overall survival. These data have been added to Figure S2 and support the negligible contribution of thermogenesis and WAT browning to the cachexia phenotype.

We agree that the increase in spontaneous activity in cachectic mice housed at 30°C is an interesting finding and needs further study. We speculate that there is finite “pool” of energy available for thermogenesis and activity. At 22°C, the available energy is diverted to thermogenesis. At 30°C, thermogenesis is negligible, so the energy is free for spontaneous activity. The specific pathways that regulate spontaneous activity in this setting are unknown.

- Authors show that treatment with ActRII-Fc and Anamorelin improve cachexia, which has already been described before. These findings, therefore, lack novelty, and no effort is done to underline any conceptual or practical advance.

We agree that ActRIIB-Fc has been reported to improve muscle mass and survival in mice bearing tumor allografts like C26 and LLC (Hatakeyama et al., 2016; Nissinen et al., 2018; Toledo et al., 2016; Zhou et al., 2010). We are unaware of any study showing improved survival in cachectic mice treated with anamorelin, however the reviewer’s point is accepted given its clinical success.

In the KL model, ActRIIB-Fc alone was not enough to improve survival unless it was paired with an agent that effectively increased food intake. This conceptual innovation challenges the existing dogma of cachexia monotherapy and warrants publication in a top tier journal such as Nature Communications. Furthermore, this is the first study, to our knowledge, to reverse cachexia in a genetically engineered mouse model with pharmacologic interventions. In a field with no approved therapies, we believe this milestone is a significant leap forward.

There are also an unusually large number of formal or precise comments on inaccuracies, mistakes or unclear panels throughout the manuscript. A strong effort must be done to clarify the manuscript, as it is extremely hard to follow and draw any clear conclusion.

- Figure 1 legend: the phrasing is not clear, revise and clarify please. Also, indicate the sex of the mice in the legend and in Methods section.

Revised for clarity.

- Figure 1A: explain better what is the difference between +CACS and – CACS mice; it becomes clear as the manuscript advances, but these groups are not properly explained from the beginning. Were both models inoculated with AdCre? What is the comparison marked with the *?

Revised for clarity.

- Figure S1: describe in the legend the mouse genotypes and treatments shown, as well as the number of mice analyzed for each condition. Statistical methods described in the legend do not correspond with the panels. Also, specify the type of post-tests performed for correction of the statistical analysis, when appropriate.

We have re-designed this figure while adding the number of mice analyzed per panel and updated the statistical methods used in the legend for all panels.

- Figure S1C: it seems that -CACS mice have larger adipocytes in WAT and BAT than WT mice. Please, measure adipocyte size and quantify in each group, using a reasonable number of mice and fields to perform proper statistics.

Thank you for this excellent suggestion. We have replaced the previously scanned images (20x) for others with higher magnification (40x) so that tissue details are more apparent. We have also used automated methods and blinded investigators for the quantification of adipocyte size (Adiposoft), as well as lipid content in the fat and UCP1 staining in the BAT and WAT (QuPath). Now shown in Figure S1.

- Figure S1D: define more clearly what is exactly being shown in the bar graph and the dot graph: number of genes? Batlas Score? % of BAT genes?

We have now expanded this explanation in the main text. The BATLAS algorithm allows for the identification of gene signatures associated to brown adipose tissue, based on RNA-Seq expression data. The numbers provided by the algorithm estimates the fraction of brown and white adipose tissue content per sample. As the authors show in their paper, this can be used to estimate browning of the WAT (Perdikari et al., 2018).

- Figure S1F: quantify IHQ staining. The images are not clear, they look like +CACS mice have larger adipocytes in their BAT, in contrast with FS1C. Also, a higher magnification image of the gWATs would help identifying the UCP1-positive cells.

Thank you, we have replaced the images with high resolution 40X images that we used for automated quantification using QuPath.

- Figure S1G: it is referred to after figure S1H, which is strange. UCP1 protein levels in gWAT are missing in this figure, which is an important measurement since they seem to be highly induced in mRNA, but IHQ is not so clear. Also, from this WB, neither UCP1 nor PGC1a protein levels look decreased in BAT, at least not consistently. Please, quantify these bands relative to GAPDH (which, by the way, in a metabolically imbalanced situation as cachexia, might not be the most appropriate reference protein; please comment on this point).

The order of the panels has been updated in Figure S1. We were unable to detect UCP1 protein in the gWAT by Western blot using commercially available antibodies, so we decided to remove this panel. On page 6, line 21 we disclose this limitation. The immunohistochemistry images have now been quantified by a blinded investigator using automated methods, as described above.

- Figure S1H: what are these mRNA levels relative to? It seems that they are relative to Leptin expression in -CACS mice (value of 1), but this is never indicated. Instead of Student t test, one-way ANOVA should be used for this analysis, since there are several groups being compared. The same applies to Figure S1I and others: when more than two groups are being compared for one single parameter, one-way ANOVA and not Student t test should be used, with the appropriate correction for multiple comparisons.

We have now amended and modified the manuscript to correct this issue. We compare relative mRNA expression levels between -CACS and +CACS for each gene. Since there are only 2 groups (-CACS and +CACS) for each transcript, the use of Student's t-test is preferred, such as in the new Fig S1F.

Also, Pgc1a levels are not significantly increased in +CACS gWATs, indicate this in the main text. In this regard, in the text authors specify that “UCP1 expression as increased over 50-fold”. Since no protein quantification is shown, the mouse gene expression should be indicated by using italics and lower case (“Ucp1”). As suggested above, protein quantification from IHQ and WB would also be very welcome.

We thank the reviewer for highlighting this issue. After revisiting the statistical tests performed on that panel (t-test comparing -CACS to +CACS for each individual transcript) we found the expression to be significantly different with a p-value of 0.0473. Moreover, we also provide the quantification of UCP1 in the gWAT to support this claim.

- Figure 1D: this panel is very confusing: the different groups are scattered throughout the graph and the findings indicated in the text are not clearly reflected in it: I do not find +CACS mice being the lowest; instead, I find the 30²C mice being the lowest (both +/-CACS). +CACS at 22²C mice are scattered around, but not necessarily lower than -CACS or Pre-CACS. If anything, +CACS mice have lower total EE only when lean mass is very low, but not when lean mass is high. This reflects that really cachectic mice (with very low lean mass) have lower EE. I do not think that this graph is the clearest way to say this.

Also, what does the red line represent? All +CACS mice, or only 22²C +CACS mice? This is not clear from the legend. Are the slopes different between +CACS, -CACS and WT (indicating a stronger influence of lean mass in one of the groups)? And does temperature influence this effect/slope?

There are numerous challenges with the analysis of energy expenditure data in mice (Müller et al., 2021). The current figure is in line with recent guidelines, and we are hesitant to deviate from this standard. However, the reviewer’s criticisms are acknowledged, and we made updates to add clarity. First, we increased the sample size of CACS mice to increase the confidence in the analysis. The solid red line includes +CACS mice at 22 and 30°C since there was no significant difference between the conditions. The slopes for each line are not significantly different. Only the y-intercept has changed. Temperature had no effect on the slopes. These details have been updated in the legend.

- Figure S2B: decreased RER is not equivalent to decreased VO₂ consumption; instead, it indicates that metabolism is more reliant on fatty acid consumption than on glucose or aminoacids. A deeper explanation about this decreased RER finding and its implications should be included in the main text when mentioning this finding.

We have now expanded on this explanation on page 8 of the text.

- Figure 1E and 1F: how are food intake and activity in WT mice? What time points are significantly different between groups? 2-way ANOVA tests also give differences for individual points, not only for the whole curve as indicated in the figure.

Please see below for the food intake and activity in a cohort of mice that included WT compared to -CACS and +CACS. We decided to compare whole curves instead of individual points using 2-way ANOVA given the obvious differences between the groups and the desire to keep the figures uncluttered.

Figure 12. Cumulative Activity and Food intake in KL and WT mice. Cumulative activity measured in meters(A) and Cumulative food intake in Kcal for WT (non-induced mice, n=12), - CACS (n = 11) and +CACS(n = 9). Comparissons were performed using 2-way ANOVA (*p = 0.05)

Also for Figure 1F, it is surprising that mice at 30°C move more than mice at 22°C, I would have expected the opposite, at least for the -CACS mice. Please comment on this finding.

Indeed, the increase in spontaneous activity at 30°C is interesting. This phenomenon has been previously observed (Škop et al., 2020). Mice at typical housing temperatures (20°C–22°C) live below thermoneutrality, and about half of their total energy expenditure is devoted to maintaining core body temperature. We speculate that the reduced requirement for thermogenesis at 30°C makes more energy available for spontaneous activity but the mechanisms controlling spontaneous activity in mice are unknown. These concepts are now described in the Discussion on page 18 line 7.

- Figure 1: it is not clear what statistical test is performed for each panel. In the figure legend authors indicate that a 2-way ANOVA test was performed for panels B/D/E, but these graphs represent different types of data and 2-way ANOVA is not appropriate for panel B. Please revise and indicate accordingly.

Revised.

- Figure S4A and S4B: I understand that the slope of the linear regressions shown in panels A and B are significantly different to 0 in the glycolytic, fast-twitch gastrocnemius, quadriceps and EDL muscles, but no in the slow-twitch, less glycolytic soleus muscle, where $p > 0.05$. However, this is not explained in the text: authors do not mention which are supposed to be the glycolytic/non-glycolytic muscle types or what the slopes are representing. Without this explanation, figures are very hard to follow. Please explain these results in more detail in the text and legends.

Added to page 9, line 15.

- Figure 2B: both LDHA and PDH are glycolytic enzymes that convert pyruvate into lactate (LDH) or Acetyl-CoA (PDH). I understand that PDH drives oxidative phosphorylation for pyruvate, while LDHA drives non-respiratory glycolysis, but no explanation is given in the text, and why LDHA activity is increased in +CACS mice (more pLDHA), while PDHE1a activity is decreased (more pPDHE1a). Please explain this crucial point properly in the text. Also, please quantify these Western blots, normalizing for non-phosphorylated proteins and Tubulin, and provide a statistical analysis of these results.

We now provide more details in the text on page 9 under the section heading, **Mice with CACS Have Impaired Skeletal Muscle Metabolism**. We have also quantified the Western blots and added the results to Figure 2.

Figure 13. Western blot quantification from Figure 2. (B) Western blot of phosphorylated (Tyr10) and total LDHA, phosphorylated (Ser293) and total PDHe1 α , and Tubulin from Soleus and EDL lysates from -CACS and +CACS mice (C, D)Quantification by densitometry of blots in B. Comparisons were made using Student's t-test compared with -CACS mice: *P <0.05

What happens with these glycolysis/phosphorylation markers in other glycolytic muscles like quadriceps and gastrocnemius (larger than EDL)?

We evaluated phosphorylation of LDHA (Ser293) and PDHe1 α (S293) in gastrocnemius tissue and found them to be increased only in the cachectic female mice. We suspect that the observed differences in signaling between the EDL and gastrocnemius are due to presence of slow twitch, oxidative fibers in the gastrocnemius, but this hypothesis has not been formally addressed.

Figure 14. Western blot of LDHA (Ser293) and PDHe1 α (S293) in gastrocnemius. (A) Western blot of phosphorylated (Tyr10) and total LDHA, phosphorylated (Ser293) and total PDHe1 α , and Tubulin from gastrocnemius lysates from -CACS and +CACS mice (B, C)Quantification by densitometry of blots in A. Comparisons were made using Student's t-test compared with -CACS mice: *P <0.05

- Figure S5C: by eye, it seems that EDL from +CACS mice have more mitochondria than -CACS mice; while the opposite occurs in the soleus. However, authors report no alterations in the ultrastructural level. A quantification of the surface and evaluation of the morphology of mitochondria would be welcome to support this claim, because the shown EM photographs are not consistent with the text.

In an attempt to simplify the manuscript and reduce the amount of data included, we have decided to remove the EM images.

- Figure S5D and E: please, explain more clearly what do the codes of the different conditions mean, as they are now explained in the legend I cannot understand them. For example: "NADH-linked (N) LEAK (L) respiration was measured in the presence of pyruvate and malate (PML or PMP)", but I cannot see N or L codes in the figure. Does PM mean "NADH-linked lead respiration"? Does PML mean "NADH-linked leak respiration in the presence of pyruvate (and, if so, why the "L" subindex)?"

This legend has been revised for clarity:

"Oxygen consumption rates for permeabilized fibers obtained from Soleus (F), and EDL (G) of -CACS (n=14) and +CACS (n=19) mice. Oxygen flux due to mitochondria leak is denoted by subscript "L", NADH-linked flux is denoted by "P", and maximal flux in the presence of the mitochondrial uncoupler FCCP is denoted by "E". These rates were measured in the presence of pyruvate and malate (PM) and absence of ADP, and PM with the addition of ADP, glutamate and succinate (PMGS). Maximal oxygen flux through complex IV (CIV_E) was measured in the presence of TMPD and ascorbate."

- Figure 2 and S5: there is a shift between different glycolytic muscle types throughout the figure: soleus and EDL for Figure 2 panels A, B, C, D; gastrocnemius in G, H and I; gastrocnemius in Figure S5 panels A and G; soleus and EDL in panels B-F. Why this inconsistency in the glycolytic muscles used? Justify in the text, or include the analysis for the same type of muscle for all panels.

In our previous study (Goncalves et al., 2018) and in Fig S3A and B, we show that the muscles most affected by CACS are those rich in type II fibers such as the Extensor Digitorum Longus (EDL) and gastrocnemius. In contrast, muscles rich in type I fibers such as the soleus are spared. Given these findings and the similarity in mass and anatomic location of these two muscles, their direct comparison is a reasonable approach, in our opinion. The EDL and Soleus mass ranges from 5-8 milligrams so it is not possible to use this tissue for applications that require a large amount of biomass. In these cases, we make use of the gastrocnemius (>150 mg tissue) that contains a large proportion of type II fibers and is affected by cachexia (Fig S3A-B). However, we acknowledge that the response to cachexia is not the same in all muscles.

- Figure 3 and S6: it is not clear how the energy intake in +CACS mice (~8kcal/day) was determined. Also, this energy intake should be normalized to total body mass, because +CACS mice weight less and, therefore, subjecting a heavy, -CACS mouse to this drastic CR can be much more stressful than doing it to a thin +CACS mouse.

We apologize for the lack of clarity.

The cumulative daily food intake for mice with CACS (~8 kcal/day) is presented in Figure 1F.

The WT mice used in the CR experiment were the same age and from the same line as the KL mice that are induced with tumors. The starting body weights were matched. As expected, the implementation of CR led to the same average weight loss as observed with CACS. The decision to not normalize energy intake to total body mass was based on published guidelines (Butler and Kozak, 2010; Müller et al., 2021).

- Figure 3D: the units are not represented in the figure. Corrected

- Figure S6C: as in Figure S2B, an explanation of the metabolic implications of the reduced RER in CR mice should be included in the text. Also, there is no statistical difference, however, it is indicated in the manuscript.

RER is an indirect readout of whole-body substrate oxidation that is influenced by numerous exogenous and endogenous factors such as: the level of feeding (positive vs. negative energy balance), the size of the glycogen stores, insulin sensitivity, and the amount of lipid in the adipose tissue (Schutz, 1995). The cachectic mice at 22°C are in a state of negative energy balance with depleted white adipose tissue depots, defective hepatic fatty acid oxidation, and active glucose oxidation in the brown adipose tissue for thermogenesis. The non-cachectic mice at 30°C are in neutral energy balance with normal adipose tissue reserve, normal hepatic fatty acid oxidation, and no excessive glucose oxidation needed for thermogenesis. Since this discussion is overly complex and does not contribute to the main conclusions of the study, we have not included it in the main text.

- Figure S8C and D: it is very hard to assess most findings described in the text by the images shown in these figures, (reduced fiber size or UCP1 staining), since the magnification is too low and there is not enough detail. Also, to robustly support these descriptions, a quantification of several fields from several mice of each group, with appropriate statistics, should be performed.

We agree with the reviewer. All microphotographs (20X) were replaced for others with higher resolution and magnification(40X). We have also quantified muscle fiber area, and performed appropriate statistical tests on the results. For the gWAT and BAT H&E and UCP1 staining we have performed whole slide supervised automated quantification using QuPath.

- Figure 4: authors affirm that the combined therapy protects against the loss of glycolytic skeletal muscle, however, the increase in the amount of these tissues is very low. Please, include data of -CACS mice to verify if this protection is comparable to -CACS mice.

We have rephrased the text to more accurately describe our findings: “The combination therapy led to a subtle improvement in lean mass but not fat mass (Figs. 5C, D), and had no effect on overall survival, tumor burden, spontaneous activity, RER, or EE (Figs. 5E, S5A-D)”. Additionally, we have added a panel

F

Figure 15. Food intake in mice with CACS from Figure 1F

with % change of Fat mass and Lean Mass, measured by EchoMRI of all the mice in the study (Fig 5C and D). We believe this measure provides a more accurate representation of the changes in whole body lean mass. As the reviewer noted, this is still a subtle change for the GDF intervention with a p-value of 0.06 after a one-way ANOVA with Tukey multiple comparisons test.

- “Anamorelin hydrochloride” has been approved as a treatment for CACS. Why is “anamorelin fumarate” (Ana) used in the present manuscript? If authors are trying to compare “Anamorelin fumarate” against an already approved treatment, please indicate, or discuss this difference.

Both anamorelin hydrochloride and anamorelin fumarate provide the same free base equivalent, and anamorelin fumarate was the more accessible and cost-effective option.

- Figure 4A and B: statistics should be performed using the one-way ANOVA (4 groups, one parameter), not the indicated two-way ANOVA. Are +CACS mice significantly different from -CACS mice in these serum parameters?

We agree, and this issue is now corrected.

Also, when describing the antibody treatments, please explain what “QW, SQ” means.

Added to methods.

- Figure 4D: it is extremely striking that the difference in weight loss between start and end of treatments is significant in the combination group (where some mice increase and others decrease their weight), while it is not significant in the other groups, with more consistent and stronger changes. Please revise this statistical assessment.

We agree and apologize for this error. All the data and statistical approaches for each panel have been reviewed in detail with a statistician. The figures and legends have been updated.

- In the S11A and S11B panels, authors indicate that the Ana + ActRIIB-Fc treatment significantly increases food intake and weight loss after two weeks. However, the animals treated only with Ana (a drug already used against CACS) did not show any increase compared to the IgG group. Is the Ana dose and regimen used appropriate in this model? Please comment.

The dose of 30 mg/kg P.O. that has been reported several times in the literature for xenograft models (Bernardo et al., 2020; Northrup et al., 2013; Pietra et al., 2014). We have now added these references to the methods on page 37. We agree with the reviewer that the synergy observed between the ActRIIB-Fc treatment and anamorelin in this model is novel and intriguing. We think it highlights the poorly understood mechanisms of anorexia and the heterogeneity of cachexia amongst animal models.

Figure S11H: please, specify the meaning of the different signs for statistical significance: * and #. The figure legend is not clear.

Symbols have all been defined in the legends.

- Figure S12: it is not clear how many male mice were analyzed. In many figures, only one dot is represented (C, D, E). Please, specify how many male mice are included in this assay. If they were too few, as it seems from some of the graphs, the negative conclusion reached by authors for male mice are not robust enough and should be commented in the text and Discussion section.

For the GDF15 trial, 39 mice (22 males, 17 females) were induced, and 29 mice (17 males, 12 females) reached 15% weight loss and underwent randomization. For the Ana cohort, a total of 50 mice (26 males, 24 females) were induced and 46 mice (23 males, 23 females) reached 15% weight loss and underwent randomization. The pre-specified primary outcome was the percent weight loss at 2 weeks following the start of treatment (End). Secondary outcomes included food intake, overall survival, body composition, food intake, spontaneous activity, skeletal muscle mass, and white adipose tissue mass. Given the heterogenous timing of weight loss and death at the individual level with the KL model, not all mice were available for each secondary endpoint. The sample size for each secondary measure is included in the legends. These details have been added to the “**Therapeutic trials**” section of the methods.

- Figure 5G: authors claim that combination therapy in females increased overall survival; but there is no sign indicating significance in this graph. If combination therapy-treated mice live longer, indicate significance; otherwise, restate these finding in the text (both in Results and in Discussion), indicating a “clear tendency” towards longer lifespan.

We have now amended this in Fig 5J and added the result of the analysis using Log-rank Mantel-Cox test.

Methods:

- Please, indicate the sequences of the primers used for qPCRs.

Only Ppargc1a, RER1 and Rpl41 were amplified with SYBR and we have added the primer sequence to Table3. These sequences were blasted to guarantee the specificity of the amplification product and had been previously validated by others(Perez et al., 2017). The rest of the transcripts were measured using Taqman Probes.

- Describe how the CACS- and CACS + animals have been identified.

This has been now added to the methods section.

- The Caloric Restriction experiments carried out in this manuscript are not described in the methods section.

We apologize for this omission. We have now added this information to the methods section in page 39 under “Caloric restriction experiments”.

REFERENCES

- Abreu-Vieira, G., Xiao, C., Gavrilova, O., and Reitman, M.L. (2015). Integration of body temperature into the analysis of energy expenditure in the mouse. *Molecular Metabolism* 4, 461-470.
- Bernardo, B., Joaquim, S., Garren, J., Boucher, M., Houle, C., LaCarubba, B., Qiao, S., Wu, Z., Esquejo, R.M., Peloquin, M., et al. (2020). Characterization of cachexia in the human fibrosarcoma HT-1080 mouse tumour model. *J Cachexia Sarcopenia Muscle* 11, 1813-1829.
- Butler, A.A., and Kozak, L.P. (2010). A recurring problem with the analysis of energy expenditure in genetic models expressing lean and obese phenotypes. *Diabetes* 59, 323-329.
- Fearon, K., Strasser, F., Anker, S.D., Bosaeus, I., Bruera, E., Fainsinger, R.L., Jatoi, A., Loprinzi, C., MacDonald, N., Mantovani, G., et al. (2011). Definition and classification of cancer cachexia: an international consensus. *The Lancet Oncology* 12, 489-495.
- Goncalves, M.D., Hwang, S.-K., Pauli, C., Murphy, C.J., Cheng, Z., Hopkins, B.D., Wu, D., Loughran, R.M., Emerling, B.M., Zhang, G., et al. (2018). Fenofibrate prevents skeletal muscle loss in mice with lung cancer. *Proc Natl Acad Sci U S A* 115, E743-E752.
- Lehtio, J., Arslan, T., Siavelis, I., Pan, Y., Socciarelli, F., Berkovska, O., Umer, H.M., Mermelekas, G., Pirmoradian, M., Jonsson, M., et al. (2021). Proteogenomics of non-small cell lung cancer reveals molecular subtypes associated with specific therapeutic targets and immune evasion mechanisms. *Nat Cancer* 2, 1224-1242.
- Müller, T.D., Klingenspor, M., and Tschöp, M.H. (2021). Revisiting energy expenditure: how to correct mouse metabolic rate for body mass. *Nat Metab* 3, 1134-1136.
- Northrup, R., Kuroda, K., Duus, E.M., Barnes, S.R., Cheatham, L., Wiley, T., and Pietra, C. (2013). Effect of ghrelin and anamorelin (ONO-7643), a selective ghrelin receptor agonist, on tumor growth in a lung cancer mouse xenograft model. *Support Care Cancer* 21, 2409-2415.
- Perez, L.J., Rios, L., Trivedi, P., D'Souza, K., Cowie, A., Nzirorera, C., Webster, D., Brunt, K., Legare, J.F., Hassan, A., et al. (2017). Validation of optimal reference genes for quantitative real time PCR in muscle and adipose tissue for obesity and diabetes research. *Sci Rep* 7, 3612.
- Pietra, C., Takeda, Y., Tazawa-Ogata, N., Minami, M., Yuanfeng, X., Duus, E.M., and Northrup, R. (2014). Anamorelin HCl (ONO-7643), a novel ghrelin receptor agonist, for the treatment of cancer anorexia-cachexia syndrome: preclinical profile. *J Cachexia Sarcopenia Muscle* 5, 329-337.
- Pocock, S.J. (2013). *Clinical Trials*. (John Wiley & Sons Ltd.).
- Rogier, L.C.P., Guido, H., Renger, F.W., and Klaske van, N. (2021). Relevance of cancer cachexia models – muscle whole genome gene expression in human and animal cachexia. *Research Square*.
- Schutz, Y. (1995). Abnormalities of fuel utilization as predisposing to the development of obesity in humans. *Obes Res* 3 *Suppl* 2, 173s-178s.
- Škop, V., Guo, J., Liu, N., Xiao, C., Hall, K.D., Gavrilova, O., and Reitman, M.L. (2020). Mouse Thermoregulation: Introducing the Concept of the Thermoneutral Point. *Cell reports* 31, 107501107501.

Tschop, M.H., Speakman, J.R., Arch, J.R., Auwerx, J., Bruning, J.C., Chan, L., Eckel, R.H., Farese, R.V., Jr., Galgani, J.E., Hambly, C., et al. (2011). A guide to analysis of mouse energy metabolism. *Nat Methods* 9, 57-63.

Van Klinken, J.B., van den Berg, S.A.A., Havekes, L.M., and Willems Van Dijk, K. (2012). Estimation of activity related energy expenditure and resting metabolic rate in freely moving mice from indirect calorimetry data. *PLoS One* 7, e36162-e36162.

REVIEWER COMMENTS

Reviewer #1 (Remarks to the Author):

The authors have addressed all critical issues sufficiently and greatly improved the manuscript.

Reviewer #2 (Remarks to the Author):

The authors have appropriately addressed the concerns raised in the review, and I have no further comments.

Daniel L. Marks

Reviewer #3 (Remarks to the Author):

The submission is a revision of a manuscript that carefully characterizes a preclinical lung cancer model of cachexia that expands to examine sex and several pharmacological treatments. There are several strengths, including the comprehensive characterization of the preclinical model, comparison to caloric restriction, a novel cachexia treatment design, and the examination of sex.

The authors provided an expansive response to prior concerns in their rebuttal. However, in the revised manuscript, most prior concerns were inadequately addressed. Therefore, the authors were viewed to be minimally responsive to previous concerns. Weaknesses remain with the weak premise in the introduction, the generalized overview of cancer cachexia, the addressing of anorexia, a lack of premise for the interesting treatment design, the lack of premise for sex effects, the lack of a testable hypothesis, critical data being in the supplement, lack of clarity in many aspects of the results some specific comments are below. Overall a stronger premise that leads to a more focused presentation of the novel treatment design and sex effects would significantly enhance the predicted impact of the study.

Major

Introduction: The cancer anorexia cachexia syndrome (CACS) is a highly prevalent wasting
Three syndromes associated with progressive loss of skeletal muscle and adipose tissues that
4 predicts increased chemotherapy toxicity, complications from cancer surgery, and
overall mortality.

Comment: For clarity for the cachexia field and the readers, please define and contrast your CACS with the more general term cancer cachexia and describe the prevalence of CACS with different types of cancer cachexia. Please address that an accepted definition of cachexia is that it can not be reversed by nutritional support. At the minimum, this should be addressed in the introduction, and the relationship to anorexia cachexia explained.

Introduction Paragraph 2 Comment: Please rewrite paragraph 2 to reflect lung cancer and/or your specific preclinical model. The generalization to all cancers is implied and is not the current state of understanding for many in the field of cancer cachexia.

Minor

Previous comments mentioned the addition of a hypothesis in the introduction. Below is included in the introduction, page 5, line 41 and is not an actual hypothesis. Please revise.

>> we hypothesized that anorexia and hypermetabolism would contribute to negative energy balance in CACS and used the KL model to interrogate the changes in food intake, peripheral organ metabolism, and EE that occur following the induction of lung cancer.

Abstract: It is unclear whether this net reduction in energy is due solely to anorexia or if a combination

of anorexia and increased energy expenditure (EE) occurs.

Comment: Please rephrase this vague statement or remove it. There is considerable research on the role of cachexia; at a minimum, this is specific to the type of preclinical cancer model, most very well characterized, and the type and stage of cancer in the human.

Figure 1A is challenging to examine and verges on uninterpretable. Please add mean BW changes for each group with dot blots similar to 1B.

Figure 1B presents uncorrected tumor mass (mg)

Figure 1E, please present if the linear regressions were significant, please present statistics indicating significance or rephrase

>> Interestingly, we 122 observed a stepwise reduction in TEE with WT values being the highest and CACS 123 (+CACS) being the lowest. Pre-CACS and tumor-bearing weight stable (-CACS) mice 124 had intermediate values.

The critical data related to the results discussion of adipose tissue browning and uncoupling is in the supplement. Related to previous comments, please add relevant browning data to the figures in the manuscript if it is critical or remove it.

Reviewer #4 (Remarks to the Author):

I want to thank the authors for their efforts in addressing my comments and those from other reviewers. I think the manuscript is much clearer and robust now.

Most of my comments have been properly addressed. I still have some minor doubts and comments:

- [R4] Authors show that treatment with ActRII-Fc and Anamorelin improve cachexia, which has already been described before. These findings, therefore, lack novelty, and no effort is done to underline any conceptual or practical advance.

[Authors] We agree that ActRIIB-Fc has been reported to improve muscle mass and survival in mice bearing tumor allografts like C26 and LLC (Hatakeyama et al., 2016; Nissinen et al., 2018; Toledo et al., 2016; Zhou et al., 2010). We are unaware of any study showing improved survival in cachexic mice treated with anamorelin, however the reviewer's point is accepted given its clinical success. In the KL model, ActRIIB-Fc alone was not enough to improve survival unless it was paired with an agent that effectively increased food intake. This conceptual innovation challenges the existing dogma of cachexia monotherapy and warrants publication in a top tier journal such as Nature Communications. Furthermore, this is the first study, to our knowledge, to reverse cachexia in a genetically engineered mouse model with pharmacologic interventions. In a field with no approved therapies, we believe this milestone is a significant leap forward.

[R4] I would appreciate an explicit comment on the novelty of the combined therapy approach in the preclinical and clinical area, specially given the Anamorelin use for CACS patients, and to the relevant translational application of these findings. I cannot find any strong reference in the text to the relevance of this advance in the field.

- [R4] Figure S1D (new Figure S1H): define more clearly what is exactly being shown in the bar graph and the dot graph: number of genes? Batlas Score? % of BAT genes?

[Authors] We have now expanded this explanation in the main text. The BATLAS algorithm allows for the identification of gene signatures associated to brown adipose tissue, based on RNA-Seq expression data. The numbers provided by the algorithm estimates the fraction of brown and white adipose tissue content per sample. As the authors show in their paper, this can be used to estimate browning of the WAT (Perdikari et al., 2018).

[R4] I understand the general concept of the BATLAS analysis; but I still do not see the meaning of each of the two panels: are they representing the same parameter, such as BATLAS score? What is the difference between the two panels, and what do they exactly represent?

- [R4] Figure 3 and S6: it is not clear how the energy intake in +CACS mice (~8kcal/day) was determined. Also, this energy intake should be normalized to total body mass, because +CACS mice weight less and, therefore, subjecting a heavy, -CACS mouse to this drastic CR can be much more stressful than doing it to a thin +CACS mouse.

[Authors] We apologize for the lack of clarity.

[Authors] The cumulative daily food intake for mice with CACS (~8 kcal/day) is presented in Figure 1F.

[R4] Please, indicate the origin of this value in Figure 1F in the text; as it is, readers cannot know where this value was obtained from.

[Authors] The WT mice used in the CR experiment were the same age and from the same line as the KL mice that are induced with tumors. The starting body weights were matched. As expected, the implementation of CR led to the same average weight loss as observed with CACS.

[R4] There is no graph comparing WT and CACS mice to prove that CR led to the same average body weight loss as observed with CACS. Figure 1A is very hard to compare with Figure 3A. Maybe a line in Figure 3A including the CACS mice from Figure 1A could help here.

[Authors] The decision to not normalize energy intake to total body mass was based on published guidelines (Butler and Kozak, 2010; Müller et al., 2021).

[R4] These references are very well taken, and I admit a mistake in my comment. Indeed, data should always be normalized to lean body mass, not to total body mass. Since lean body mass is also affected in CACS mice, and following the cited references, data should be normalized to lean mass.

- Table 2 is mis-referenced in the text (page 6 line 75); and there is no reference in the text to present Table 1.

- Figure S3A-B: the Mass values are extremely high, could it be that the correct units are mg?

- References to Figures from S5 and S6 are shuffled in the text, it is rather confusing. I would place them in the order they are referred to. Also, I think there are a few mistakes in references to panels, such as that made in Page 16 line 275, talking about female mice but referring to panels from male mice. Please revise.

General remark:

- Please, when including asterisks to compare two sets of data in a panel with several sets, indicate clearly what precise two sets of data are being compared.

- Some Figures presented in the response to reviewers are not included in the manuscript, and I think they would provide valuable information: Figure 12 and 14, for example.

REVIEWER COMMENTS

Reviewer #1 (Remarks to the Author):

The authors have addressed all critical issues sufficiently and greatly improved the manuscript.

Reviewer #2 (Remarks to the Author):

The authors have appropriately addressed the concerns raised in the review, and I have no further comments.

Daniel L. Marks

Reviewer #3 (Remarks to the Author):

Comment: For clarity for the cachexia field and the readers, please define and contrast your CACS with the more general term cancer cachexia and describe the prevalence of CACS with different types of cancer cachexia. Please address that an accepted definition of cachexia is that it can not be reversed by nutritional support. At the minimum, this should be addressed in the introduction, and the relationship to anorexia cachexia explained.

Thank you for the opportunity to clarify the difference between CACS and “cancer cachexia.” Both have been used interchangeably in the literature for over 30 years to describe the clinical entity currently defined as unintended weight loss >5% loss over the prior 6 months or >2% weight loss with BMI <20 kg/m². We agree that cancer cachexia, as currently defined, cannot be reversed by nutritional support. However, a large proportion of patients with lung-cancer induced weight loss also experience anorexia, which can be reversed by nutritional support. Therefore, we selected the term CACS to highlight the contributions of both anorexia and cachexia to the weight loss observed in the KL model.

We have revised the abstract and introduction to more appropriately introduce cachexia, anorexia, and the term CACS.

Introduction Paragraph 2 Comment: Please rewrite paragraph 2 to reflect lung cancer and/or your specific preclinical model. The generalization to all cancers is implied and is not the current state of understanding for many in the field of cancer cachexia.

We have revised the introduction to more appropriately introduce cachexia and anorexia with a focus on patients with lung cancer and the most common pre-clinical lung cancer cachexia model, LLC.

Minor

Previous comments mentioned the addition of a hypothesis in the introduction. Below is included in the introduction, page 5, line 41 and is not an actual hypothesis. Please revise.

>> we hypothesized that anorexia and hypermetabolism would contribute to negative energy balance in CACS and used the KL model to interrogate the changes in food intake, peripheral organ metabolism, and EE that occur following the induction of lung cancer.

We have revised the hypothesis statement to be more in-line our approach. “we hypothesized that CACS could be reversed by targeting both anorexia and Activin A-induced metabolic dysfunction.”

Abstract: It is unclear whether this net reduction in energy is due solely to anorexia or if a combination of anorexia and increased energy expenditure (EE) occurs.

Comment: Please rephrase this vague statement or remove it. There is considerable research on the role of cachexia; at a minimum, this is specific to the type of preclinical cancer model, most very well characterized, and the type and stage of cancer in the human.

The sentence has been removed.

Figure 1A is challenging to examine and verges on uninterpretable. Please add mean BW changes for each group with dot blots similar to 1B.

Figure 1A contains individual data points demonstrating the progression of weight loss for the KL model. We believe there is value to the reader to see the variability in the onset of CACS in this model, which is in contrast to the more homogenous allograft models commonly used in the field. However, the reviewer's critique is valid and we have now added a new panel (Figure 1B), depicting the 'weight normalized to peak' at endpoint for each mouse. Thank you for this suggestion.

B

Figure 1B presents uncorrected tumor mass (mg).

Figure 1B (new Figure 1C) represents total lung mass, a surrogate for tumor burden in this model. For clarity, we changed the y axis to "Lung mass" to prevent any confusion.

Figure 1E, please present if the linear regressions were significant, please present statistics indicating significance or rephrase

>> Interestingly, we 122 observed a stepwise reduction in TEE with WT values being the highest and CACS 123 (+CACS) being the lowest. Pre-CACS and tumor-bearing weight stable (-CACS) mice 124 had intermediate values.

In order to appropriately compare the different groups, we normalized TEE to lean mass using ANCOVA following the recommendations of the National Mouse Metabolic Phenotyping Center (MMPC, www.mmpc.org) and other leaders in this field^{1,2}. The data has been added to a new panel (Figure 1G) and statistical testing indicators have been applied.

G

The critical data related to the results discussion of adipose tissue browning and uncoupling is in the supplement. Related

to previous comments, please add relevant browning data to the figures in the manuscript if it is critical or remove it.

This data was an important part of the phenotyping we performed and led us to further examine the energy expenditure of the mice. When we found that the energy expenditure was low, the browning was thought to be non-contributory to the phenotype. Therefore, the data is not showcased in the main figures but still important enough to keep in the supplement. Also, other reviewers showed interest in these findings during the first round of reviews and we believe there is value for the community to see these results. We discussed this issue with the journal editor.

Reviewer #4 (Remarks to the Author):

I want to thank the authors for their efforts in addressing my comments and those from other reviewers. I think the manuscript is much clearer and robust now.

Most of my comments have been properly addressed. I still have some minor doubts and comments:

Thank you for your continued effort to improve our manuscript.

[R4] I would appreciate an explicit comment on the novelty of the combined therapy approach in the preclinical and clinical area, specially given the Anamorelin use for CACS patients, and to the relevant translational application of these findings. I cannot find any strong reference in the text to the relevance of this advance in the field.

We thank the reviewer for highlighting the relevance of our work. We have now revised the discussion to make this important point more evident. Please see the paragraphs starting on page 21 line 10, page 22 line 1, and page 23 line 1.

[R4] I understand the general concept of the BATLAS analysis; but I still do not see the meaning of each of the two panels: are they representing the same parameter, such as BATLAS score? What is the difference between the two panels, and what do they exactly represent?

We completely agree with the reviewer on the redundancy of the panels. They are both representing the same information that is the BAT/WAT ratio estimated by BATLAS. We have removed the horizontal bar graph for simplicity.

[R4] Please, indicate the origin of this value in Figure 1F in the text; as it is, readers cannot know where this value was obtained from.

On page 12, line 10 we now state: “To assess the contribution of reduced food intake to TEE and skeletal muscle metabolism, we performed an experiment where WT mice were calorie-restricted (CR) to consume the same energy as mice with CACS (8 kcal/day as shown in Fig 1H).”

[R4] There is no graph comparing WT and CACS mice to prove that CR led to the same average body weight loss as observed with CACS. Figure 1A is very hard to compare with Figure 3A. Maybe a line in Figure 3A including the CACS mice from Figure 1A could help here.

We agree with the reviewer’s assessment. In order to better make this comparison, we have included a new panel (Figure 1B) that shows the weight normalized to peak of the mice at endpoint. In addition, we have added a dashed line to Figure 3A to indicate the average weight loss of the CACS mice from Figure 1B so that it can be directly compared to the CR protocol.

[R4] These references are very well taken, and I admit a mistake in my comment. Indeed, data should always be normalized to lean body mass, not to total body mass. Since lean body mass is also affected in CACS mice, and following the cited references, data should be normalized to lean mass.

Thank you for this suggestion. In order to appropriately compare the different groups, we normalized TEE to lean mass using ANCOVA following the recommendations of the National Mouse Metabolic Phenotyping Center (MMPC, www.mmpc.org) and other leaders in this field^{1,2}. The data has been added to a new panel (Figure 1G) and statistical testing indicators have been applied.

- Table 2 is mis-referenced in the text (page 6 line 75); and there is no reference in the text to present Table 1.
Corrected

- Figure S3A-B: the Mass values are extremely high, could it be that the correct units are mg?
Indeed. We apologize for the typo and corrected the mistake.

- References to Figures from S5 and S6 are shuffled in the text, it is rather confusing. I would place them in the order they are referred to. Also, I think there are a few mistakes in references to panels, such as that made in Page 16 line 275, talking about female mice but referring to panels from male mice. Please revise.

We reviewed the figure references and corrected all identified mistakes.

We agree that splitting the discussion of Figure S5 is not ideal. However, the current structure allows the reader to directly compare the effects of anti-GDF15 against Ana for lung mass, activity, RER, and EE on the same figure. The structure

mirrors the data presented in Figure 5. If the reviewer would like, we could move panels S5 E-H to S6 or a completely new supplemental file during the publication processing.

General remark:

- Please, when including asterisks to compare two sets of data in a panel with several sets, indicate clearly what precise two sets of data are being compared.

Thank you. In the new version of the manuscript, we have made an effort to make the comparisons explicit by adding bars connecting the variables that are being compared.

- Some Figures presented in the response to reviewers are not included in the manuscript, and I think they would provide valuable information: Figure 12 and 14, for example.

Thank you for this suggestion. During both rounds of revisions, it was recommended that we cut down on the amount of data presented and simplify the conclusions of the manuscript. Therefore, we felt that adding the cumulative activity and food intake from WT mice would further complicate the figure and distract from the main message of the paper.

In the case of the WB shown in Rebuttal Figure 14, we agree that it is interesting to note differences in the metabolic response of muscles with varying fiber composition. However, we believe the large proportion of type I fibers in the medial gastrocnemius may be masking the metabolic deficits that occur in the type II fibers. The soleus and EDL comparisons help to better distinguish the differences because EDL is nearly all type II fibers.

References.

- 1 Fernandez-Verdejo, R., Ravussin, E., Speakman, J. R. & Galgani, J. E. Progress and challenges in analyzing rodent energy expenditure. *Nat Methods* **16**, 797-799, doi:10.1038/s41592-019-0513-9 (2019).
- 2 Tschop, M. H. *et al.* A guide to analysis of mouse energy metabolism. *Nat Methods* **9**, 57-63, doi:10.1038/nmeth.1806 (2011).

REVIEWERS' COMMENTS

Reviewer #4 (Remarks to the Author):

Authors have addressed all my comments properly. I think the manuscript is clearer now, thank you for your effort.